# KrwEmd: Revising the Imperfect Recall Abstraction from Forgetting Everything

## Abstract

A recent research has shown that an extreme interpretation of imperfect recall abstraction – completely forgetting all past information – has led to excessive abstraction issues. Currently, there are no hand abstraction algorithms that effectively integrate historical information. This paper aims to develop the first such algorithm. Initially, we introduce the KRWI abstraction for Texas Hold'em-style games, which categorizes hands based on K-recall winrate features that incorporate historical information. Statistical results indicate that, in terms of the number of distinct infosets identified, KRWI significantly outperforms POI, an abstraction that identifies the most abstracted infosets that forget all historical information. Following this, we introduce the KrwEmd algorithm, the first hand abstraction algorithm to effectively use historical information by combining K-recall win rate features and earth mover's distance for hand classification. Experimental studies conducted in the Numeral211 Hold'em environment show that under identical abstracted infoset sizes, KrwEmd not only surpasses POI but also outperforms state-of-the-art hand abstraction algorithms such as Ehs and PaEmd. These findings suggest that incorporating historical information can significantly enhance the performance of hand abstraction algorithms, positioning KrwEmd as a promising approach for advancing strategic computation in large-scale adversarial games.

## 1 Introduction

Imperfect recall abstraction has proven to be very important for solving large-scale computational games, significantly reducing computational complexity. Recently, AI using imperfect recall abstraction has developed better-than-human strategies for Texas Hold'em testbed—even when using limited computational resources [23, 7, 8].

The task of hand abstraction in Texas Hold'em aims to reduce computational overhead by applying the same strategy to similar hands. In an imperfect recall setting [29, 20], the hand abstraction in the later phase does not strict depend on the results of the hand abstraction in the earlier phase. However, the term **imperfect recall** is often interpreted in an extreme manner in practice. Researchers typically understand it as completely forgetting all past information—in other words, considering only future information—and design abstraction algorithms based on this understanding [16, 17, 19, 15, 14]. There are two major factors that mainly affect the results of abstraction for each phase: the number of clustering centers (i.e. centroids), which can be set man-

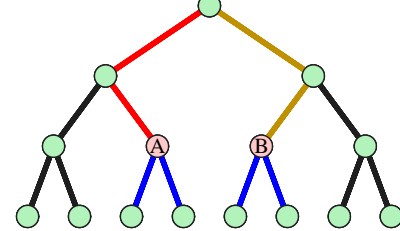

Figure 1: In a 4-phase game hand abstraction task, the current goal is to classify hands A and B.

Submitted to 38th Conference on Neural Information Processing Systems (NeurIPS 2024). Do not distribute.

ually, and the number of distinct features that are used to categorize hands at each phase. Recent research [12] has found that constructing hand features solely based on future information can lead to excessive abstraction. For example, as shown in the Figure 1, two hands: A and B constructed with only future information can have the same hand features. As the game progresses, the rate of feature repetition among different hands gradually increases, while the distribution of distinct hand features assumes a spindle-shaped pattern. Additionally, constructing hand features with historical information in addition to the future may differentiate two hands sharing the same future information and hence makes more features available for clustering as well as enhances the performance of hand abstraction.

However, there still remain two unsolved issues. First, Fu et al. [12] have introduced a K-recall outcome feature, which incorporates historical information. This feature can only identify if elements are identical or not, but it lacks the capability to discern the extent of differences between features. Therefore, it is difficult to adjust the number of clusters appropriately, which makes it challenging to construct an effective hand abstraction algorithm that integrates historical information. Second, due to the inability to modify the number of clusters, Fu et al. [12] only compared the performance between the maximum clusters cases of integration of historical information (KROI) and no integration at all (POI). In this condition, although KROI significantly outperforms POI, the comparison is inconclusive because KROI recognizes more abstracted infosets than POI. Thus, it does not prove that the performance of abstraction algorithms that integrate historical information is necessarily superior under the condition of having the same number of abstracted infosets.

This paper introduces a framework for constructing hand features based on winrates, with the K-recall winrate feature being the most crucial one. Based on this, we developed the K-recall winrate isomorphism (KRWI), an abstraction that integrates historical information. Across the same game phases, KRWI identifies slightly fewer hand features than KROI but significantly more than POI. Importantly, the K-recall winrate feature is capable of discerning the extent of differences between features. Therefore, by combining the earth mover's distance with the K-recall winrate feature, we developed the first hand abstraction algorithm that integrates historical information, named KrwEmd, and designed an efficient computational method. We validated our approach in the Numeral211 game environment, where KrwEmd demonstrated superior performance to POI under the same infosets conditions. Additionally, in clustering settings, KrwEmd also outperformed the Ehs and PaEmd algorithms, with PaEmd being the current state-of-the-art hand abstraction algorithm.

## 2   Background and Notation

Generally, Texas Hold'em-style poker games are modeled as imperfect information games. However, for the task of hand abstraction, games with ordered signals [18, 12] offer a better theoretical tool. The game with ordered signals is a subclass of imperfect information games in that they further subdivide the nodes (also called histories, states, or trajectories) in imperfect information games into mutually independent signals and public nodes. This allows for each aspect to be studied in isolation. Under this framework, the hand abstraction task in Texas Hold'em-style games is modeled as signal abstraction.

In a game with ordered signals $\tilde{\Gamma} = \left\langle \tilde{\mathcal{N}}, \tilde{H}, \tilde{Z}, \tilde{\rho}, \tilde{A}, \tilde{\chi}, \tilde{\tau}, \gamma, \Theta, \varsigma, O, \omega, \succeq, \tilde{u} \right\rangle$, there is a set of players $\tilde{\mathcal{N}} = \mathcal{N} \cup \{c, pub\}$, which includes not only the main participants $\mathcal{N} = \{1, \ldots, N\}$ but also a special nature player $c$ who controls the randomness and an observer player $pub$ who can see everything but doesn't take any actions. The game progresses through a series of public nodes $\tilde{X} = \tilde{H} \cup \tilde{Z}$. Some of these public nodes are terminal public nodes $\tilde{Z}$ where the game ends and outcomes are determined, while the others are non-terminal public nodes $\tilde{H}$. Among the non-terminal public nodes, some are where players make decisions within the action space $\tilde{A}$, and the remaining are chance public nodes where the nature player reveals signals, with the special action $Reveal$ within $\tilde{A}$.

At every non-terminal public node, $\tilde{\rho} : \tilde{H} \mapsto \mathcal{N}c$ (i.e., $\mathcal{N} \cup \{c\}$) specifies which player is responsible for making an action, and $\tilde{\chi} : \tilde{H} \mapsto 2^{\tilde{A}}$ confines the possible actions they can take. When the nature player makes a move, it reveals signals $\theta \in \Theta$ that carry information relevant to the game. These signals are then observed by all players except $c$, $O(\theta) = (O_1(\theta), \ldots, O_N(\theta), O_{pub}(\theta))$, though what they can see might differ.

The progression from one public node to another is clearly defined $\tilde{\tau} : \tilde{H} \times \tilde{A} \mapsto \tilde{X}$, ensuring that the game's structure is sequential and predictable. Similarly, the signals are revealed according to a probability distribution $\varsigma : \Theta \mapsto \Delta(\Theta)$, which specifies the likelihood of the next signal given the current one. We use $\tilde{h} \sqsubseteq \tilde{h}'$ to indicate that $\tilde{h}$ is a predecessor of $\tilde{h}'$, and $\theta \sqsubseteq \theta'$ to indicate that $\theta$ is a predecessor of $\theta'$. Each phase of the game is the number of times nature player has revealed signals, denoted by $\gamma : \tilde{X} \mapsto \mathbb{N}^+$. $\mathfrak{r} = \{\gamma(\tilde{x}) \mid \tilde{x} \in \tilde{X}\}$ represents the phases that a game with ordered signals may go through. Since the root is a chance public node, we have $\min \mathfrak{r} = 1$.

At the end of the game, players receive their payoffs based on the signals and the terminal public node, represented by $\tilde{u} = (\tilde{u}_1, \ldots, \tilde{u}_N)$, where $\tilde{u}_i : \Theta \times \tilde{Z} \mapsto \mathbb{R}$. Additionally, each player's survival status is determined at these terminal public nodes, denoted by $\omega = (\omega_1, \ldots, \omega_N)$, where $\omega_i : \tilde{Z} \mapsto \{true, false\}$. The signals possess a partial order within their subset, terminal signals $\tilde{\Theta}$, indicated by $\succeq : \tilde{\Theta} \times \mathcal{N} \times \mathcal{N} \mapsto \{true, false\}$. It is required that for any terminal signal $\theta \in \tilde{\Theta}$ and terminal public nodes $\tilde{z} \in \{\tilde{z}' \in \tilde{Z} \mid \omega_i(\tilde{z}') = \omega_j(\tilde{z}') = true\}$, if $\succeq (\theta, i, j) = true$, then $\tilde{u}_i(\theta, \tilde{z}) \geq \tilde{u}_j(\theta, \tilde{z})$.

Players make decisions based on their observations of signals and the current non-terminal public node. A player may have the same observation for different signals, forming a signal infoset for signals they cannot distinguish. For a player $i \in \mathcal{N}$, the signal infoset for a signal $\theta$ is denoted as $\vartheta_i(\theta) = \{\theta' \in \Theta \mid O_i(\theta) = O_i(\theta') \wedge O_{pub}(\theta) = O_{pub}(\theta')\}$. Specifically, for the nature player, $\vartheta_c(\theta) = \{\theta' \in \Theta \mid O_{pub}(\theta') = O_{pub}(\theta)\}$. We abuse the notation $\vartheta \in \Theta_i$ to represent a signal infoset, where for any player $i \in \mathcal{N}$, $\Theta_i$ is a partition of $\Theta$, representing the collection of player $i$'s signal infosets. $\Theta_i^{(1)}, \ldots, \Theta_i^{(|\mathfrak{r}|)}$ are the collections of player $i$'s signal infosets for each phase, and they form a partition of $\Theta_i$. In games with ordered signals, the signals describe all private information. The signal infoset, combined with public nodes, can be transformed into the infoset of an imperfect information game. Fu et al. [12] detailed this transformation process.

The game with ordered signals model allows us to study the issue of signal abstraction independently. For this purpose, we introduce a signal (infoset) abstraction profile, $\alpha = (\alpha_1, ., \alpha_N)$, where for each player $i \in \mathcal{N}$, $\alpha_i$ is a partition of $\Theta$ called the signal (infoset) abstraction. Any $\hat{\vartheta} \in \alpha_i$ then is said to be an abstracted signal infoset for player $i$, and it can be further divided into several signal infosets within $\Theta_i$. These finer signal infosets collectively form a partition of $\hat{\vartheta}$. In general, two signal abstractions cannot be directly compared in terms of performance, but in a few specific cases there does exist a special relationship between them, which is called refinement. Consider two abstractions $\alpha_i$ and $\beta_i$. If $\forall \hat{\vartheta} \in \beta_i$, there exists one or more abstracted signal infosets in $\alpha_i$ such that the union of these forms a partition of $\hat{\vartheta}$, then we said that $\alpha_i$ refines $\beta_i$, symbolically $\alpha_i \sqsupseteq \beta_i$. The signal abstracted game $\tilde{\Gamma}^\alpha$ was derived by substituting $\Theta_i$ with $\alpha_i$ across all $\tilde{x} \in \tilde{X}$.

Perfect/imperfect recall originally describes a property of imperfect information games, indicating that players do not need to remember all the information they have observed throughout the game. Since games with ordered signals are a subset of imperfect information games, we derived the concept of signal perfect/imperfect recall from them. A player $i$ in a game $\tilde{\Gamma}$ is said to have signal perfect recall if, for any $\theta_1', \theta_2' \in \vartheta'$, any predecessor $\theta_1$ of $\theta_1'$ has a corresponding predecessor $\theta_2$ of $\theta_2'$ such that $\theta_2 \in \vartheta(\theta_1)$. If all players have signal perfect recall, the game $\tilde{\Gamma}$ is said to have signal perfect recall. For a game $\tilde{\Gamma}$ with signal perfect recall, if $\alpha_i$ is the signal abstraction of player $i \in \mathcal{N}$, let $(\alpha_i, \Theta_{-i})$ denote the signal abstraction profile where player $i$ adopts the signal abstraction $\alpha_i$ while other players do not do abstraction. If $\tilde{\Gamma}^{(\alpha_i, \Theta_{-i})}$ retains signal perfect recall, then $\alpha_i$ is considered a signal abstraction with perfect recall; otherwise, it is an signal abstraction with imperfect recall.

In games with ordered signals, the strategy $\pi_i$ for player $i$ maps from a non-terminal public node and a signal infoset to a probability distribution over actions, with the strategy profile denoted as $\pi = (\pi_1, \ldots, \pi_N)$. When all players adopt the strategy profile $\pi$, the expected sum of future rewards, also known as expected value, for player $i$ at public node $\tilde{x}$ and signal $\theta$ is denoted as $v_i^\pi(\theta, \tilde{x})$, and the expected value for the entire game is denoted as $v_i(\pi)$. A Nash equilibrium is a strategy profile where no player can obtain a higher expected value by changing their strategy. Formally, $\pi^*$ is a Nash equilibrium if for every player $i$, $v_i(\pi^*) = \max_{\pi_i} v_i(\pi_i, \pi_{-i}^*)$, where $\pi_{-i}$ denotes the strategies of all players except $i$. In two-player zero-sum scenarios, the exploitability of $\pi$ is denoted as $\epsilon(\pi) = \frac{\max_{\pi_1'} v_i(\pi_1', \pi_2) + \max_{\pi_2'} v_i(\pi_1, \pi_2')}{2}$.

## 3 Related Work

Our research focuses on hand abstraction techniques in AI systems for Texas Hold'em-style games (i.e. the signal abstraction in games with ordered signals), building on the initial works of Shi and Littman [25] and Billings et al. [4]. These seminal works introduced the concept of game abstraction, which aims to simplify games while preserving essential characteristics. The researchers started by manually forming hand buckets as a result of their expertise with game-playing strategy. The first automated hand abstraction was that of Gilpin and Sandholm [16]. Later, a model of games with ordered signals was given for Texas Hold'em by Gilpin and Sandholm [18]; lossless isomorphism (LI) was developed with signal rotation. Despite the elegance of LI, its low compression rates hinder its application in large-scale games, whereas lossy abstraction shows potential for such application. An expectation-based clustering method was proposed by Gilpin and Sandholm [17] in their work, and a histogram-based clustering method was introduced by Gilpin et al. [19]. The former is known as Ehs, while the latter is referred to as the potential-aware method. Subsequent studies by Gilpin and Sandholm [15] and Johanson et al. [20] compared Ehs and potential-aware methods, concluding that the latter holds an advantage in large-scale games. Johanson et al. [20] also introduced the use of earth mover's distance[1] (EMD) in potential-aware methods. Ganzfried and Sandholm [14] introduced a more efficient approximation algorithm for earth mover's distance in potential-aware methods (PaEmd). Brown et al. [9] further applied PaEmd to distributed environments for solving large-scale imperfect-information games. This paradigm has found success in Texas Hold'em AI systems and is considered state-of-the-art in hand abstraction. Very recently, Fu et al. [12] proposed several novel tools, such as abstraction resolution and common refinement. They introduced two signal abstraction: one is the potential outcome isomorphism (POI), which identifies the maximum number of abstracted signal infosets considering future information only; The other is the K-recall outcome isomorphism (KROI), which identifies the maximum number of abstracted signal infosets considering historical information. They emphasized that current imperfect recall signal abstraction algorithms, which consider only future information, are prone to excessive abstraction. However, they did not provide practical signal abstraction algorithms.

Other abstraction techniques for decision-making problems include action abstraction [13, 6, 21] and general imperfect recall abstraction [10, 11] in extensive-form games, as well as state abstraction and action abstraction in reinforcement learning [1, 2].

## 4 Winrate Isomorphism

The first contribution of this paper is an isomorphism framework of winrate-based features, including the potential winrate isomorphism (PWI) and the k-recall winrate Isomorphism (KRWI). Compared with outcome-based features, winrate-based features offer a streamlined approach, focusing exclusively on the distribution of loss, draw, and win outcomes of signals emanating from a signal infoset (and its predecessors) as it evolves towards the terminal signals. Winrate-based features are numerical vectors of consistent length. In this section, an identical Winrate-based feature uniquely determines an abstracted signal infoset. It is worth noting that the similarity of Winrate-based features reflects the similarity among signal infosets, allowing for clustering based on these features (see Section 5).

Both PWI and KRWI share the similar isomorphism construction process for player $i$ in phase $r$, as illustrated in algorithm 1. The difference lies only in the construction operator for the winrate-based features, FEATURE, used in lines 5 and 12. The isomorphism construction process starts by iterating through all signal infosets of $\Theta_i^{(r)}$ and collecting their features. Next, these features are deduplicated and stored in lexicographical order within set $\mathcal{C}_i^{(r)}$, which is implemented as a vector data structure. Within $\mathcal{C}_i^{(r)}$, the index of a feature serves as an identifier for an abstracted signal infoset. Then, by utilizing a hash table $\mathcal{CI}_i^{(r)}$, we can identify an abstracted signal infoset's identifier based on its feature. In the final step, we traverse $\Theta_i^{(r)}$ again, associating the identifier of a signal infoset with the identifier of its corresponding abstracted signal infoset, and this relationship is recorded in $\mathcal{D}_i^{(r)}$, an isomorphism map. The function $Index_i(r, \cdot)$ is a domain-specific mapping that assigns a unique identifier to each signal infoset at phase $r$, within the numeric range of 0 to $|\Theta_i^{(r)}| - 1$. In

---

[1]https://en.wikipedia.org/wiki/Earth_mover%27s_distance

---

**Algorithm 1** Isomorphism Constructor

---

**Require:**

    $r = 1, \ldots, R$. Phases.

    $\Theta_i^{(r)}$. Signal infoset space for player $i$.

    $Index_i(r, \cdot) : \Theta_i^{(r)} \mapsto \mathbb{N}$. Signal infoset index function for player $i$.

  1: **procedure** ISOMORPHISMCONSTRUCTOR($r, \Theta_i^{(r)}$, FEATURE($\cdot$))

  2:      Initialize $\mathcal{C}_i^{(r)}$ vector as empty.

  3:      Initialize $\mathcal{D}_i^{(r)}$ array arbitrarily with length $|\Theta_i^{(r)}|$.

  4:      **for** $\vartheta \in \Theta_i^{(r)}$ **do**

  5:          $feature \leftarrow$ FEATURE($\vartheta$).

  6:          Append $feature$ to $\mathcal{C}_i^{(r)}$.

  7:      **end for**

  8:      Eliminate duplicates from $\mathcal{C}_i^{(r)}$.

  9:      Sort the elements of $\mathcal{C}_i^{(r)}$ in lexicographical order.

10:      Construct hash table $\mathcal{CI}_i^{(r)}$ from $\mathcal{C}_i^{(r)}$. Store the index $lexid$ and value $feature$ of $\mathcal{C}_i^{(r)}$ in $\mathcal{CI}_i^{(r)}$ as key-value pairs $(feature, lexid)$.

11:      **for** $\vartheta \in \Theta_i^{(r)}$ **do**

12:          $feature \leftarrow$ FEATURE($\vartheta$), $idx \leftarrow Index_i(r, \vartheta)$.

13:          Update $\mathcal{D}_i^{(r)}[idx]$ with $\mathcal{CI}_i^{(r)}[feature]$.

14:      **end for**

15:      **return** $(\mathcal{C}_i^{(r)}, \mathcal{D}_i^{(r)})$.

16: **end procedure**

---

Texas Hold'em-style games, one optional approach for implementing this function is through lossless isomorphism [18, 27].

## 4.1 Potential Winrate Isomorphism

Potential winrate isomorphism (PWI) is a signal abstraction that classify signal infosets based on its potential winrate features. These features focus on the distribution of a player's winrate over terminal signals after passing through a given signal infoset, without considering the history of how the player reached the signal infoset. Specifically, for player $i$ in phase $r$, the potential winrate feature associated with $\vartheta \in \Theta_i^{(r)}$ is defined as

$$pf_i^{(r)}(\vartheta) = (pf_i^{(r),0}(\vartheta), pf_i^{(r),1}(\vartheta), \ldots, pf_i^{(r),N}(\vartheta)), \tag{1}$$

where

- $pf_i^{(r),0}(\vartheta)$ denotes the probability that player $i$ ranks lower than least one other player in the terminal signals, after passing through $\vartheta$.

- $pf_i^{(r),l}(\vartheta)$, for $l > 0$, denotes the probability that player $i$ ranks no lower than any other player and ranks higher than exactly $l - 1$ other players in the terminal signals, after passing through $\vartheta$.

In the terminal phase, the winrate feature is calculated by directly statisticing the game outcomes for players in the given signal infoset. Moreover, in the non-terminal phases, we use a recursive approach to simplify the computation of the winrate feature, thereby avoiding the need to enumerate every signal infoset down to the terminal phase. The recursive formula is

$$pf_i^{(r),l}(\vartheta) = \sum_{\substack{\vartheta^{(r+1)} \in \Theta_i^{(r+1)} \\ \vartheta \sqsubseteq \vartheta^{(r+1)}}} pf_i^{(r+1),l}(\vartheta^{(r+1)}) Pr\{\vartheta^{(r+1)} | \vartheta\} \tag{2}$$

| | Preflop | Flop | | Turn | | | River | | | |
|---|---|---|---|---|---|---|---|---|---|---|
| Recall | 0 | 0 | 1 | 0 | 1 | 2 | 0 | 1 | 2 | 3 |
| KRWI | 169 | 1028325 | 1123442 | 1850624 | 34845952 | 37659309 | 20687 | 33117469 | 529890863 | 577366243 |
| KROI | 100 | 1137132 | 1241210 | 2337912 | 38938975 | 42040233 | 20687 | 39792212 | 586622784 | 638585633 |
| W/O (%) | 100.0 | 90.43 | 90.51 | 79.16 | 89.49 | 89.58 | 100.0 | 83.23 | 90.33 | 90.41 |

Table 1: The number of abstracted signal infosets identified by KRWI, and KROI in each phase and $k$ of HUNL&HUNLE, with W/O indicating the ratio identified by PWI and POI.

The PWI algorithm is derived from the POI algorithm [12], and the details of the PWI algorithm are elaborated in Appendix A.1. Both algorithms use the potential winrate feature to distinguish between different abstracted signal infosets in the terminal phase. However, unlike POI, PWI also uses the potential winrate feature in non-terminal phases to identify different abstracted signal infoset classes, while POI relies on the potential outcome

| | Preflop | Flop | Turn | River |
|---|---|---|---|---|
| LI | 169 | 1286792 | 55190538 | 2428287420 |
| PWI | 169 | 1028325 | 1850624 | 20687 |
| POI | 169 | 1137132 | 2337912 | 20687 |
| W/O (%) | 100.0 | 90.43 | 79.16 | 100.0 |

Figure 2: The number of abstracted signal infosets identified by LI, PWI, and POI in each phase of HUNL&HUNLE, with W/O indicating the ratio identified by PWI and POI.

feature (which captures the distribution of the abstracted signal infoset class for future signal infoset). In non-terminal phases, the potential winrate feature is a simplified version of the potential outcome feature. Unsurprisingly, PWI also results in excessive abstraction similar to POI. As shown in Table 2, in heads-up limit hold'em (HULHE) and heads-up no-limit hold'em (HUNL), the number of abstracted signal infosets identifiable by lossless isomorphism increases with each phase, indicating that the game becomes increasingly complex. However, the number of abstracted signal infosets identifiable by PWI and POI first increases and then decreases, showing a spindle-shaped pattern. And we observed that when only future information is considered, winrate-based features may lead to greater information loss compared to outcome-based features. For instance, in the River phase, the number of abstracted signal infosets identified by PWI is only 79.16% of that identified by POI.

## 4.2 K-Recall Winrate Isomorphism

As Fu et al. [12] mentioned, supplementing historical information can enhance the ability of signal abstraction to identify abstracted signal infosets. Inspired by KROI's construction approach, we developed the k-recall winrate isomorphism (KRWI). The key difference is that instead of using k-recall outcome features to distinguish between different signal infosets, KRWI utilizes k-recall winrate features.

In a game with signal perfect recall, all signals within the signal infoset $\vartheta$ have their predecessors at phase $r'$, which belong to the identical signal infoset $\vartheta'$. For player $i$ at phase $r$, the signal infoset $\vartheta \in \Theta_i^{(r)}$ has a $k$-recall winrate feature ($k < r$) represented as a numerical array with a dimension of $(k+1)(N+1)$:

$$rf_i^{(r,k)}(\vartheta) = (pf_i^{(r)}(\vartheta); pf_i^{(r-1)}(\vartheta); \ldots; pf_i^{(r-k)}(\vartheta)) \tag{3}$$

When $r'$ is less than $r$, $pf_i^{(r')}(\vartheta)$ denotes the potential winrate feature for the predecessor signal infoset $\vartheta'$ of $\vartheta$ at phase $r'$. Since we have stored all the potential winrate features of $\vartheta \in \Theta_i^{(r)}$ through $\mathcal{PC}_i^{(r)}, \mathcal{PD}_i^{(r)}$ and assigned them unique identifiers in Algorithm A1. To save storage space and facilitate retrieval, what we actually store is

$$rfi_i^{(r,k)}(\vartheta) = (\mathcal{PD}_i^{(r)}[\vartheta], \mathcal{PD}_i^{(r-1)}[\vartheta], \ldots, \mathcal{PD}_i^{(r-k)}[\vartheta]) \tag{4}$$

$\mathcal{PD}_i^{(r')}[\vartheta]$ is the identifier for the potential winrate feature of the predecessor $\vartheta'$ of $\vartheta$ in the $r'$ phase, $r' \leq r$. For algorithm details, please refer to Appendix A.2.

Just as the potential winrate feature is a simplified version of the potential outcome feature, the k-recall winrate feature is a simplified version of the k-recall outcome feature. Table 1 shows the number of signal infosets that KRWI and KROI can identify and their ratio in HUNL&HULHE. We were pleasantly surprised to find that while the ratio of PWI to POI resolution can drop below 80%,

when $k$ is set to its maximum value, i.e. $r - 1$, the ratio of KRWI to KROI resolution can reach nearly 90% at a minimum, with most of the information preserved. Also, we can easily observe that the number of abstracted signal infosets identified by KRWI is much higher than that identified by POI.

## 5 K-Recall Winrate Abstraction with Earth Mover's Distance

Fu et al. [12] introduced potential and k-recall outcome features, referred to as outcome-based features, to distinguish different abstracted signal infosets. In the previous section, we developed potential and k-recall winrate features, termed winrate-based features, for the same purpose. In these two methods, Each unique feature corresponds to a single abstracted signal infoset. Intuitively, we can infer that feature similarity might reflect the similarity among abstracted signal infosets, enabling further abstraction and compression for application in large-scale games. However, assessing similarity with outcome-based features is challenging because the identification code indicates only the category, without reflecting the degree of similarity. In contrast, winrate-based features represent winrates, which are inherently comparable, allowing for an easy definition of distances between them.

For the signal information sets $\vartheta, \vartheta'$ of player $i$ at phase $r$, we can define the distance of their k-recall winrate feature as

$$d(rf_i^{(r,k)}(\vartheta), rf_i^{(r,k)}(\vartheta')) = \sum_{j=0}^{k} w_j \cdot \text{Emd}(pf_i^{(r-j)}(\vartheta), pf_i^{(r-j)}(\vartheta')) \quad (5)$$

Among Equation (5), Emd is the operator used to calculate the earth mover's distance (EMD) [24]. The EMD calculates the distance between two histograms using optimal transport theory. Since it requires solving linear programming equations, the computational complexity of the EMD is sensitive to the dimensionality of the histograms, and approximate algorithms are usually used for larger-scale problems. However, the dimensionality of winrate-based features is small, with a dimension of 3 in a two-player scenario, so we attempt to use a fast algorithm for accurately computing the EMD [5]. $w_0, \ldots, w_k$ are hyperparameters used to control the importance of EMD at each phase $r, \ldots, r - k$. We use the KMeans++ algorithm [3], combined with the distance of their k-recall winrate feature, to cluster the abstracted signal infosets of KRWI. We named this algorithm KrwEmd.

Although calculating EMD on small-dimensional histograms is already very fast, clustering actual Texas Hold'em still faces a significant computation. For example, for the River phase of HUNL&HULHE, the clustering input size of the KRWI abstracted signal infoset is approximately $5.8 \times 10^8$. When we set the number of centroids to 20000, a single Kmeans++ iteration takes about 19000 core hours on a computer with a 2.40GHz clock frequency, which is a significant time cost. Therefore, we need to find ways to reduce this time cost. We have developed an accelerated algorithm, please refer to Appendix A.3 for details.

## 6 Experimental Setup

We conducted experiments on the Numeral211 Hold'em [12] testbed. Numeral211 is a two-player three-phase Taxes Hold'em-style game with more complex hand systems than the Leduc Hold'em [26] and Rhode Island Hold'em [25] test environments, making it suitable for studying hand abstraction issues. Detailed rules are included in Appendix B. Table 3 shows the number of abstracted signal infosets recognized by KRWI and KROI, along with lossless isomorphism, in Numeral211 Hold'em.

| | Preflop | Flop | | Turn | | |
|---|---|---|---|---|---|---|
| LI | 100 | 2260 | | 62020 | | |
| Recall | 0 | 0 | 1 | 0 | 1 | 2 |
| KRWI | 100 | 2234 | 2248 | 3957 | 51000 | 51070 |
| KROI | 100 | 2250 | 2260 | 3957 | 51176 | 51228 |
| W/O (%) | 100.0 | 99.29 | 99.47 | 100.0 | 99.67 | 99.69 |

Figure 3: The number of abstracted signal infosets identified by LI, PWI, and POI in each phase of HUNL&HUNLE, with W/O indicating the ratio identified by PWI and POI.

Let $\alpha = (\alpha_1, \alpha_2)$ be the signal abstraction we would like to assess. We will test the strength of the signal abstraction by measuring exploitability of the approximate equilibrium derived using the

299 CSMCCFR algorithm [30, 22] in different abstracted signal infoset scales. We gauge the performance
300 over exploitability. For doing that, we consider both symmetric and asymmetric abstraction scenarios.

301 In this symmetric abstraction setting, we measure the exploitability of approximate equilibrium
302 that is yielded when both the players in the game employ signal abstraction in the original game.
303 However, it may lead to the abstraction pathology [28]. To avoid such problems, we illustrate the
304 theoretical performance of the signal abstraction under evaluation through asymmetric abstraction.
305 The approximate equilibrium in the signal abstracted games $\tilde{\Gamma}^{(\alpha_1,\Theta_2)}$ and $\tilde{\Gamma}^{(\Theta_1,\alpha_2)}$ is obtained to
306 obtain $\pi^{*,1}$ and $\pi^{*,2}$, respectively. Finally, we concat the two strategies to get $\pi' = (\pi_1^{*,1}, \pi_2^{*,2})$ and
307 check the exploitability of $\pi'$.

## 7 Experiment

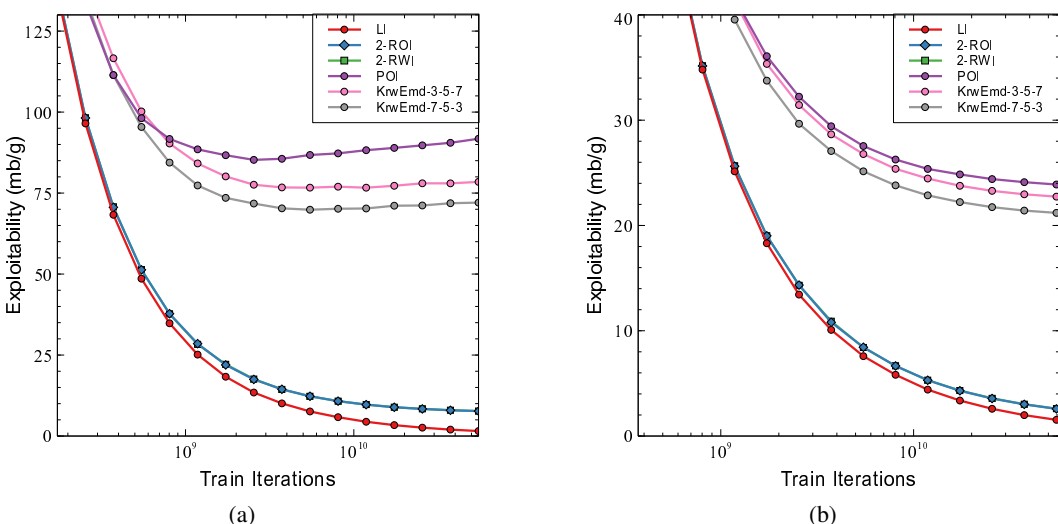

Figure 4: Full abstraction setting experiment, trained for $5.5 \times 10^{10}$ iterations.

309 Firstly, we provide an evaluation of the performance of KRWI (2-RWI) compared with KROI (2-ROI)
310 and POI (0-ROI) approaches and lossless isomorphism. We keep the most abstracted signal infosets
311 identified under the full abstraction setting. Note that POI is the common refinement of existing
312 signal abstraction algorithms that only consider future information. And, since previous works cannot
313 control the number of abstracted infoset, they cannot justify their performance in that considering
314 historical information in signal abstraction was better than that in signal abstraction with the same
315 number of abstracted infoset. To investigate this issue, we included KrwEmd and set the clustering
316 scale to be consistent with POI. Note here, that 2-RWI and 2-ROI share the same capability of infoset
317 recognition in Preflop and Flop, while POI is only a little bit worse than 2-RWI and 2-ROI in Flop.
318 Thus, we can directly allow clustering of KrwEmd abstraction use the abstracted signal infosets
319 identified by POI in Preflop and Flop, and only perform clustering in River. Here, we design four
320 sets of hyper-parameters: $(w_0, w_1, w_2)$, i.e., exponentially decreasing: $(16, 4, 1)$, linearly decreasing:
321 $(7, 5, 3)$, constant: $(1, 1, 1)$, and increasing: $(3, 5, 7)$ in the importance of historical information. We
322 only show the result of best- and worst-performing parameters (to make the figure neat). The full
323 figures appear in the Appendix C. Figure 4a shows the result of symmetric abstraction, while Figure
324 4b shows the result of asymmetric abstraction. We observed that both symmetric and asymmetric
325 abstractions maintained consistent abstraction performance without abstraction pathologies. As
326 expected, overfitting was observed in the symmetric abstraction scenario while in the asymmetric
327 scenario overfitting was significant only for POI. The performance difference between 2-RWI and
328 2-ROI is small, which means that under the full abstraction setting, using simple winrate-based
329 features instead of complex outcome-based features can achieve nearly the same performance. Even
330 with the worst parameter configuration (increasing importance), KrwEmd with the same number of
331 abstracted signal inforsets as POI still outperforms POI.

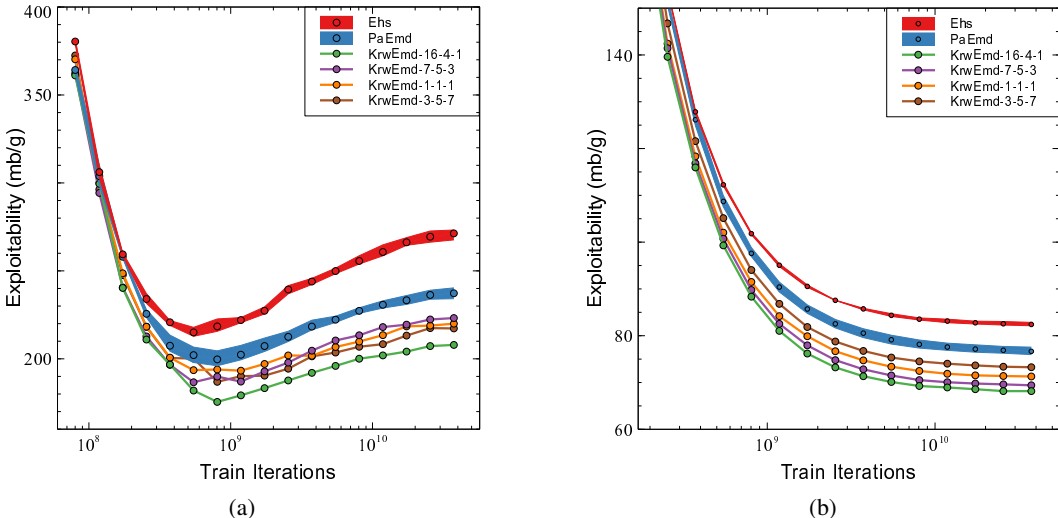

Figure 5: Performance comparison of KrwEmd versus other imperfect recall signal abstraction algorithms considering only future information, trained for $3.7 \times 10^{10}$ iterations.

Next, we compared the performance of KrwEmd with the currently applied signal abstraction algorithms Ehs and PaEmd. It should be noted that POI is the common refinement both for Ehs and PaEmd, meaning that the maximum number of abstracted signal infosets they can recognize will not exceed that of POI. Thus, we set a compression rate that is 10 times lower than that of POI, while not performing abstraction for Preflop. The final number of abstracted infosets is set to $(100, 225, 396)$. To exclude the influence of random events on performance, we generated 3 sets of abstractions for Ehs and PaEmd each. KrwEmd used hyperparameters $(w_{3,0}, w_{3,1}, w_{3,2}; w_{2,0}, w_{2,1})$ in Flop and River, which are exponentially decreasing $(16, 4, 1; 4, 1)$, linearly decreasing $(7, 5, 3; 5, 3)$, constant $(1, 1, 1; 1, 1)$, and increasing $(3, 5, 7; 5, 7)$ in the importance of historical information. Additionally, since PaEmd uses approximate EMD calculations, its approximate distance is asymmetric, making it difficult for the algorithm to converge. We truncated after 1000 iterations on a single core, with an average cost of 1427.7s, while Ehs and KrwEmd both achieved convergent clustering results, requiring an average of 12.3 and 96.7 iterations, with average time costs of 11.2s and 341.4s, respectively.

Figure 5a shows the results of symmetric abstraction experiments, while Figure 5b shows the results of asymmetric abstraction experiments. We observed that both symmetric and asymmetric abstractions maintained consistent abstraction performance, similar to the full abstraction scenario, without significant abstraction pathologies. The experimental results show that KrwEmd's performance is far superior to that of Ehs and PaEmd under all parameter settings. Our experiments also confirmed that, despite PaEmd's convergence issues, it is indeed a more effective abstraction algorithm than Ehs. Additionally, we further validated that the importance of historical information decreases progressively from bottom to top, although this time the best-performing parameter was exponentially decreasing rather than linearly decreasing as in the previous experiment.

These two experiments validate that considering historical information is indeed more effective than considering future information only in signal abstraction even in imperfect recall setting.

# 8 Conclusion

This research introduces the first imperfect recall signal abstraction algorithm that considers historical information. This algorithm has the ability to adjust the scale of the abstracted signal infosets. Based on this, we fully verified that the imperfect recall signal abstraction and abstraction algorithms considering historical information is superior to that only considering future information. Therefore, the KrwEmd algorithm has replaced the PaEmd algorithm and become the SOTA in this field. Based on the KrwEmd algorithm, we are expected to build a stronger Texas Hold'em AI.

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

**Algorithm A1** Potential Winrate Isomorphism

**Require:**
    $r = 1, \ldots, R$. Phases.
    $\Theta_i = \bigcup_{r=1}^{R} \Theta_i^{(r)}$. Signal infoset space for player $i$.
    $Index_i(r, \cdot) : \Theta_i^{(r)} \mapsto \mathbb{N}$. Signal infoset index function for player $i$.
1:  **procedure** POTENTIALWINRATEISOMORPHISM($\Theta_i$)
2:      **for** $r = R$ to 1 **do**
3:          **if** $r == R$ **then**
4:            FEATUREFUNC $\leftarrow$ POTENTIALWINRATEFEATURELASTPHASE($\cdot$).
5:          **else**
6:            FEATUREFUNC $\leftarrow$ POTENTIALWINRATEFEATURE($\cdot, r, \mathcal{PC}_i^{(r+1)}, \mathcal{PD}_i^{(r+1)}$).
7:          **end if**
8:          $(\mathcal{PC}_i^{(r)}, \mathcal{PD}_i^{(r)}) \leftarrow$ ISOMORPHISMCONSTRUCTOR($r, \Theta_i^{(r)}$, FEATUREFUNC).
9:      **end for**
10:     **return** $(\mathcal{PC}_i^{(1)}, \mathcal{PD}_i^{(1)}), \ldots, (\mathcal{PC}_i^{(R)}, \mathcal{PD}_i^{(R)})$.
11: **end procedure**
12: **procedure** POTENTIALWINRATESFEATURELASTPHASE($\vartheta$)
13:     **return** $pf_i^{(R)}(\vartheta)$                       $\triangleright$ compute according Equation (1)
14: **end procedure**
15: **procedure** POTENTIALWINRATEFEATURE($\vartheta, r, \mathcal{PC}_i^{(r+1)}, \mathcal{PD}_i^{(r+1)}$)
16:    $feature_\vartheta \leftarrow$ zero array with length $N + 1$
17:    **for** $\vartheta' \in \Theta_i^{(r+1)}$, such that $\exists \theta' \in \vartheta', \exists \theta \in \vartheta: \varsigma(\theta'|\theta) > 0$ **do**
18:      $idx \leftarrow Index_i(r+1, \vartheta'), abs \leftarrow \mathcal{PD}_i^{(r+1)}[idx], feature_{\vartheta'} \leftarrow \mathcal{PC}_i^{(r+1)}[abs]$.
19:      **for** $j = 0$ to $N$ **do**
20:         $feature_\vartheta[j] \leftarrow feature_\vartheta[j] + feature_{\vartheta'}[j] Pr\{\vartheta'|\vartheta\}$
21:      **end for**
22:    **end for**
23: **end procedure**

# A   Algorithm Details

## A.1   Potential Winrate Isomorphism

Algorithm A1 describes the computation process for potential winrate isomorphism. This algorithm operates in reverse, starting from the game's final phase $R$.

## A.2   K-Recall Winrate Isomorphism

Algorithm A2 constructs the k-recall winrate isomorphism using the k-recall winrate feature. This process requires the prior construction of the potential winrate isomorphism map $\mathcal{PD}_i^{(r)}$ using Algorithm A1.

## A.3   Accelerating Distance Computing for K-Recall Winrate Features

According to Equation (5), we note that the distance calculation between a k-recall winrate isomorphism class and a centroid's k-recall winrate feature can be decomposed into k+1 pairs of potential winrate feature EMD calculations. The potential winrate feature of the hand remains unchanged, while only the potential winrate feature of the centroid changes. Decomposing the calculation into the EMDs of potential winrate features involves significantly fewer computations than directly calculating the EMD of two k-recall winrate features. Specifically, for the River phase of HUNL&HULHE, we have the compression ratio as $\frac{169+1028325+1850624+20687}{529890863} = \frac{2899805}{529890863} = 0.0054725$.

Algorithm A3 describes how we accelerate the batch EMD computation between a centroid and all KRWI classes' k-recall winrate features. It should be noted that the K-recall winrate feature involved in the calculation of the centroid in the algorithm is in the form of Equation (3), while the K-recall winrate feature in $\mathcal{RC}^{(r,k)}$ is in the form of Equation (4). This method reduced the computational

---
**Algorithm A2** K-Recall Winrate Isomorphism

---

**Require:**
    $r = 1, \ldots, R$. Phases.

    $\Theta_i^{(r)}$. Signal infoset space for player $i$.

    $Index_i(r, \cdot) : \Theta_i^{(r)} \mapsto \mathbb{N}$. Signal infoset index function for player $i$.

    $\mathcal{PD}_i^{(r)} : \mathbb{N} \mapsto \mathbb{N}$. Potential winrate isomorphism map.

1: **procedure** KRECALLWINRATEISOMORPHISM($\Theta_i, k$)
2:     **for** $r = 1$ to $R$ **do**
3:         $k' \leftarrow$ MIN$(r - 1, k)$.
4:         FEATUREFUNC $\leftarrow$ KRECALLWINRATEFEATURE$(\cdot, r, k')$.
5:         $(\mathcal{RC}_i^{(r,k')}, \mathcal{RD}_i^{(r,k')}) \leftarrow$ ISOMORPHISMCONSTRUCTOR$(r, \Theta_i^{(r)},$ FEATUREFUNC$)$.
6:     **end for**
7:     **return** $(\mathcal{RC}_i^{(1,0)}, \mathcal{RD}_i^{(1,0)}), \ldots, (\mathcal{RC}_i^{(k+1,k)}, \mathcal{RD}_i^{(k+1,k)}), \ldots, (\mathcal{RC}_i^{(R,k)}, \mathcal{RD}_i^{(R,k)})$.
8: **end procedure**
9: **procedure** KRECALLWINRATESFEATURE$(\vartheta, r, k)$
10:     initial a empty vector $feature$.
11:     **for** $s = r$ to $r - k$ **do**
12:         $\vartheta' \leftarrow$ the predecessor signal infoset of $\vartheta$ in the $s$ phase for player $i$.
13:         $idx \leftarrow Index_i(s, \vartheta')$, $abs \leftarrow \mathcal{PD}_i^{(s)}[idx]$.
14:         Append $feature$ with $abs$.
15:     **end for**
16:     **return** $feature$
17: **end procedure**

---

cost of EMD from 19000 core hours to approximately 104 core hours, which is significantly lower than the time cost of summarizing the distance for each KRWI class, which is about 524 core hours and is an unavoidable $O(1)$ cost.

The distance batch calculation for each centroid can be processed independently and distributed across tens of multi-core computer (e.g. 96-core computers), with each computer responsible for calculating the features of some centroids in one iteration, which are then aggregated. Using this technique, we can reduce an iteration to a few hours, which is acceptable for Texas Hold'em AI training.

# B   Numerall211 Hold'em Rules

Numeral211 Hold'em is played according to the following rule:

1. **Ante:** Each player antes 5 chip into the pot at the start of the hand.

2. **Hole Card:** Both players are dealt one private card face down, known as the hole card.

3. **Deck:** The deck consists of a standard poker deck, excluding the Jokers, Kings, Queens, and Jacks, resulting in a total of 40 cards. There are four suits: spades (♠), hearts (♡), clubs (♣), and diamonds (♢), each containing ten cards numbered 2 through 9, and including the ten (T) and ace (A).

4. **First Betting Phase:** Following the distribution of hole cards, a phase of betting occurs. Players can choose to check or bet, with the bet size set at 10 chips.

5. **Flop:** After the initial betting phase, a single community card, termed the flop, is revealed from the deck.

6. **Second Betting Phase:** Another phase of betting takes place after the flop, with the bet size increasing to 20 chips.

7. **Turn:** After the Second betting phase, another community card, termed the turn, is revealed from the deck.

8. **Third Betting Phase:** Another phase of betting takes place after the turn, with the bet size still set at 20 chips.

---

**Algorithm A3** Distance Batch

---

**Require:**

    $r = 1, \ldots, R$. Phases.

    $\mathcal{RC}_i^{(r,k)} : \mathbb{N} \mapsto \mathbb{N}^{k+1}$. K-recall winrate feature set.

    $\mathcal{PC}_i^{(r)} : \mathbb{N} \mapsto [0,1]^{N+1}$. Potential winrate feature set.

    $\mathcal{PD}_i^{(r)} : \mathbb{N} \mapsto \mathbb{N}$. Potential winrate isomorphism map.

    $rc = (pc^{(r)}, \ldots, pc^{(r-k)})$. K-recall winrate feature of the input centroid.

**Ensure:**

    Distances of all k-recall winrate feature with centroid.

 1: **procedure** DISTANCEBATCH($w_0, \ldots, w_k, rc, r, k$)

    Initial phase $s$ empty earth mover's distance vector $EmdDis^{(s)}$ for $s = r, \ldots, r - k$.

    Initial empty output distance vector $Dis$.

 2:    **for** $t = 0$ to $k$ **do**

 3:        **for** $pf$ in $\mathcal{PC}_i^{(s)}$ **do**

 4:            Append $EmdDis^{(r-t)}$ with Emd($pf, rc[t]$)

 5:        **end for**

 6:    **end for**

 7:    **for** $rfi$ in $\mathcal{RC}_i^{(r,k)}$ **do**

 8:        $dis \leftarrow 0$.

 9:        **for** $t = 0$ to $k$ **do**

10:            $dis \leftarrow dis + w_t * EmdDis^{(r-t)}[\mathcal{PD}_i^{(r-t)}[rfi[t]]]$.

11:        **end for**

12:        Append $Dis$ with $dis$.

13:    **end for**

    **return** $Dis$.

14: **end procedure**

---

9. **Showdown:** If neither player folds, a showdown occurs. Players reveal their cards, aiming to form the best possible hand. The player with the highest-ranked hand wins the pot. In the case of a tie, the pot is split evenly. The Table 2 show the hand ranks of Numeral211 Hold'em.

10. **Betting Options:** Throughout the game, players have options to fold, call, or raise. In each betting phase, the total sum of bets and raises is limited to a maximum of 4, with fixed bet sizes of 10 chips in the first phase and 20 chips in the last two betting phases.

## C  Supplementary Data for Experiment 1

Figure 6 show all of the result in experiment 1.

| Rank | Hand | Prob. | Description | Example |
|------|------|-------|-------------|---------|
| 1 | Straight flush | 0.00321 | 3 of cards with consecutive rank and same suit. Ties are broken by highest card. | T♠9♠8♠2♣ |
| 2 | Three of a kind | 0.01587 | 3 of cards with the same rank. Ties are broken by the card's rank. | T♠T♡T♣2♣ |
| 3 | Straight | 0.04347 | 3 of cards with consecutive rank. Ties are broken by the highest card rank. | T♠9♡8♣2♢ |
| 4 | Flush | 0.15799 | 3 of cards with the same suit. Ties are broken by the highest card rank, then second highest card rank, then third highest card rank. | T♠8♠6♠2♣ |
| 5 | Pair | 0.34065 | 2 of cards with the same rank. Ties are broken by the rank of the pair, then by the rank of the third card. | T♠T♡8♣2♢ |
| 6 | High card | 0.43881 | None of the above. Ties are broken by comparing the highest ranked card, then the second highest ranked card, and then the third highest ranked card | T♠8♡6♣2♢ |

Table 2: Hand ranks of Numeral211 Hold'em

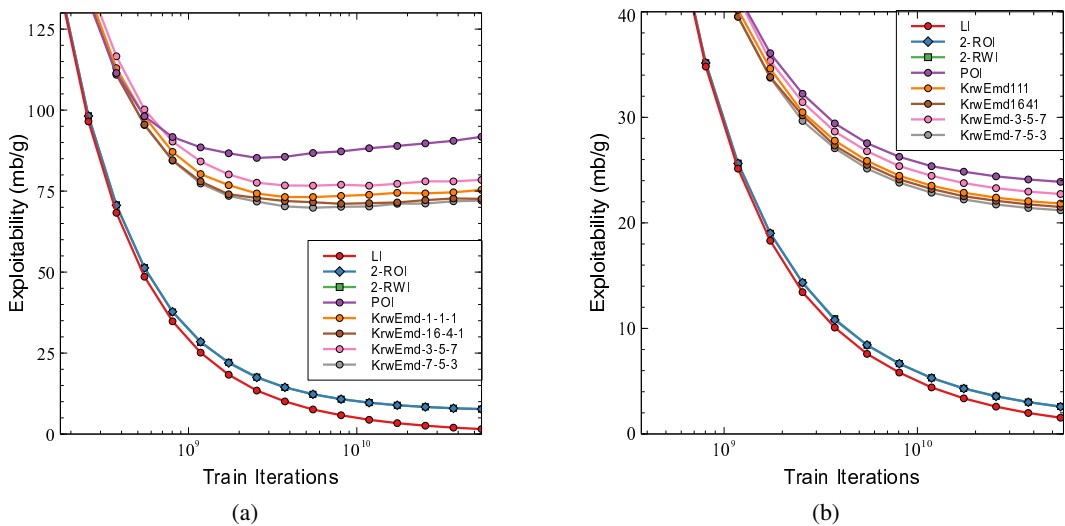

(a)        (b)

Figure 6: All data within experiment 1

