# OpenReview forum: "KrwEmd: Revising the Imperfect Recall Abstraction from Forgetting Everything"
_NeurIPS.cc/2024/Conference — Submitted to NeurIPS 2024_

### Official Review · Reviewer_HNa9 · 2024-07-13

**Soundness:** 3
**Presentation:** 3
**Contribution:** 3
**Rating:** 6
**Confidence:** 2

**Summary:**

This paper introduces the KrwEmd algorithm, a novel approach to hand abstraction in Texas Hold’em-style games. The main contribution is the integration of historical information using K-recall winrate features and earth mover’s distance, addressing the limitations of previous imperfect recall abstraction methods. The algorithm demonstrates significant performance improvements over state-of-the-art methods.

**Strengths:**

1. The integration of historical information via K-recall win rate features enhances the accuracy and reliability of hand abstraction.
2. KrwEmd significantly outperforms existing methods, showcasing its practical value and potential for real-world applications.
3. The algorithm is technically sound, with strong experimental validation supporting its claims.

**Weaknesses:**

Scalability: While the paper demonstrates KrwEmd's effectiveness in Texas Hold’em-style games, its scalability to more complex and diverse game scenarios remains to be explored.

Comparative Analysis: The paper could benefit from a more detailed comparison with existing hand abstraction methods to better highlight KrwEmd's unique advantages and limitations.

**Questions:**

1. How well does KrwEmd scale when applied to more complex and diverse game scenarios beyond Texas Hold’em? Could the authors provide any preliminary results or insights into this?

2. While KrwEmd achieves significant performance improvements, are there scenarios or tasks where additional training data might be beneficial or required? If so, what kind of training data would be most useful?

3. Are there other game models or systems that KrwEmd could enhance similarly? What challenges might arise in such integrations?

**Limitations:**

The paper primarily focuses on Texas Hold’em-style games. While this is a significant achievement, a discussion on the model's scalability and generalization to more complex and varied game scenarios would be beneficial. Including potential strategies to address these challenges would strengthen the paper.

---

> ### Author Rebuttal · Authors · 2024-08-05
>
> Thank you for your positive evaluation of our work, which has greatly encouraged us.
>
> You mentioned that we should provide a more detailed comparison with previous hand abstraction algorithms. We will adjust the hyperparameters and conduct more comparisons. The previous experiments were somewhat limited mainly because the computational cost of evaluating exploitability in the Numeral211 environment is very high, making each set of experiments take about a week. We will supplement with more experiments in the future.
>
> Below are the answers to your questions:
>
> 1. How well does KrwEmd scale when applied to more complex and diverse game scenarios beyond Texas Hold’em? Could the authors provide any preliminary results or insights into this?
>
>   Regarding the scalability of KrwEmd, we can confirm that the algorithm can be used in environments with large data volumes, such as Heads-Up No-Limit Texas Hold'em (HUNL) and Heads-Up Limit Hold'em (HULHE), as we have developed an acceleration algorithm. The main reasons for not providing a performance comparison are: 1) exploitability is not computable in HUNL & HULHE environments; 2) the computational cost of strategy solving in HUNL & HULHE is extremely high. Therefore, we validated it in a toy environment, but KrwEmd can be applied in HUNL & HULHE environments.
>
>   We are currently developing a new AI that replaces the hand abstraction module. Here is a preliminary observation: our previous AI version used a "zero-recall" or "forget everything" hand abstraction paradigm (PaEmd), which in the HUNL river phase could identify a maximum of 20,687 different categories. In contrast, using KrwEmd, the potential identifiable categories can reach up to 577,366,243, showing the significant information loss caused by previous algorithms. We plan to conduct ablation experiments on Slumbot (a public Texas Hold'em AI evaluation platform) after replacing the hand abstraction module (using the same abstraction scale configuration as before) to demonstrate the effectiveness of our algorithm. This work is currently in progress.
>
> 2. While KrwEmd achieves significant performance improvements, are there scenarios or tasks where additional training data might be beneficial or required? If so, what kind of training data would be most useful?
>
>   KrwEmd is a clustering algorithm and currently does not require additional data. However, to facilitate the computation of KrwEmd, it would be helpful to pre-store the potential winrate isomorphism (POI) and its winrates, as well as the k-recall winrate isomorphism (KRWI), in a database, rather than computing the winrates for each POI category in real-time. By adjusting the $w_j$ parameters in equation (5), we can generate KrwEmd data with different configurations.
>
> 3. Are there other game models or systems that KrwEmd could enhance similarly? What challenges might arise in such integrations?
>
>   We have not yet validated and applied our approach in environments beyond Poker, although we have tried to model the signal abstraction problem in a general way during the writing of the paper. In the future, we may look for other game environments to test the idea of "using historical information for abstraction." There are some important points to consider when finding such validation environments: 1) the game should have a mechanism to divide it into phases; 2) the game environment can be stochastic but should be static, meaning players know the state transition distribution; 3) the game should be able to extract states that are independent of other elements, such as hand cards in Texas Hold'em being independent of actions, pot size, and other elements.

---

### Official Review · Reviewer_yvNX · 2024-07-14

**Soundness:** 3
**Presentation:** 1
**Contribution:** 3
**Rating:** 5
**Confidence:** 3

**Summary:**

This paper introduces a novel approach to hand abstraction in Texas Hold'em-style poker games, addressing the limitations of current methods that often disregard historical information. The authors make two primary contributions: First, they develop KRWI (K-Recall Winrate Isomorphism), a new abstraction method that incorporates historical information from previous game phases. Second, they present KrwEmd, the first hand abstraction algorithm to effectively combine K-recall win rate features with earth mover's distance for hand classification. Through experiments conducted in the Numeral211 Hold'em environment, the authors demonstrate that KrwEmd significantly outperforms state-of-the-art algorithms such as Ehs and PaEmd in terms of exploitability, while maintaining the same number of abstracted information sets. This work shows that incorporating historical information can substantially enhance the performance of hand abstraction algorithms, potentially leading to more advanced strategic computation in large-scale adversarial games and stronger poker AI systems.

**Strengths:**

* Overall, the paper provides a new technique that is promising for an important area of research
* The results indicate strong improvements over alternative methods

**Weaknesses:**

For me, the paper's primary weakness is the presentation method. I had trouble understanding the significance and nature of the contribution from the current submission. In general, a clearer description of this area of research for people who, e.g., work on games but don't focus on poker would be quite helpful. Some specific suggestions/areas for improvement are:

* Clearer introduction of key concepts: The paper jumps into technical terms like 'imperfect recall abstraction' and 'hand abstraction' without adequately explaining them for a broader audience. A brief explanation of why these concepts are important in poker AI would be beneficial.
* More intuitive explanations of the algorithms: The descriptions of PWI, KRWI, and KrwEmd are highly technical. Including some simple examples or diagrams to illustrate how these algorithms work could greatly improve understanding.
* Better contextualization of the contribution: While the paper claims to outperform existing methods, it's not clear how significant this improvement is in the broader context of poker AI. A discussion of the practical implications of this improvement would be valuable.
* Clarification of experimental setup: The Numeral211 Hold'em environment is not well-known. A clearer explanation of how this relates to standard poker variants would help readers understand the relevance of the results.
* More accessible presentation of results: The graphs and tables are dense with information but lack clear explanations. Simplifying these visualizations or providing more guidance on how to interpret them would be helpful.
* Glossary of terms: Given the many technical terms used (e.g., 'earth mover's distance', 'K-recall winrate feature'), a glossary could be a valuable addition to help readers keep track of these concepts.

**Questions:**

Please address my comments about clarity overall and address any specifics that seem appropriate.

**Limitations:**

There's limited discussion of limitations. It would be good to include an explicit limitations section. In particular, it would be good to discuss potential computational challenges in more detail, any limitations on the scope of the evaluation, and future work that might be planned.

---

> ### Author Rebuttal · Authors · 2024-08-05
>
> Thank you for your positive evaluation of my work and for providing many constructive suggestions. These suggestions are very helpful, and we will take them seriously.
>
> You and other reviewers mentioned that this paper is somewhat obscure for researchers outside the Poker AI field. We realized that a significant reason for your appreciation of our work, yet not awarding a higher score, is due to our insufficient presentation. We accept this criticism and will improve it in future versions. We will directly incorporate and improve upon the points 1, 2, 5, and 6 you mentioned in the weakness section.
>
> For the points 3 and 4 you raised in the weakness section, we will also make improvements. We noticed that these points contain some aspects that were not clearly explained, and we will provide a brief clarification here.
>
> - Contribution of this Work to the Poker AI Community
>
>   The current most mainstream Texas Hold'em AI construction paradigm revolves around the CFR algorithm and its variants (including recent CFR algorithm variants that incorporate deep learning). Using this paradigm to construct a Texas Hold'em AI involves first determining the solution space, which means constructing an abstracted game. This step is divided into state abstraction (also known as infoset abstraction) and action abstraction, which are independent of each other. Our work falls into the category of state abstraction. After constructing the abstract game, we use the CFR algorithm in the solution space to perform strategy solving, noting that the strategy obtained at this point pertains to the abstracted game. Finally, we map the strategy from the abstracted game back to the original game for application. The construction of the abstracted game and strategy solving are independent tasks.
>
>   The poker AIs that caused a sensation between 2016 and 2019 (such as Deepstack, Libratus, and Pluribus) were primarily focused on strategy solving. In their construction of the abstracted game, they used the PaEmd abstraction algorithm for state abstraction. The PaEmd algorithm can be seamlessly replaced with KrwEmd, which could significantly improve the performance of the resulting AI, as we consider more comprehensive information. This represents a significant innovation and contribution to the field of Texas Hold'em AI construction.
>
> - Regarding the Choice of the Numeral211 Hold'em Environment
>
>   The choice of this environment was based on the following considerations: Previously, the toy environment was Flop Hold'em (which limits two-player limit Texas Hold'em to 2 phases). The issue with this environment is that with only two rounds, it cannot reflect the deficiencies of POI abstraction (the number of information sets has a spindle-shaped rather than triangular distribution). Therefore, we needed an environment with at least 3 phases.
>
>   The next problem is that in environments with at least 3 phases, the computational cost of calculating the best response is very high. To create a graph showing exploitability changes with training epochs, we needed a simpler environment. Initially, we chose a game with a deal sequence of [2,1,1] using a standard deck, but the best response calculation was still slow in this environment. One variant of poker is the Royal Deck (which only includes [T, J, Q, K, A] × [♠, ♣, ♥, ♦], a total of twenty cards). However, with this setup and a deal sequence of [2,1,1], the number of recognizable LI is very small, making k-means clustering less convincing. Therefore, we designed a custom environment with 40 cards. This environment is ideal in terms of best response calculation time and the number of recognizable LIs.

---

> ### Author Response · Authors · 2024-08-14
> **Thanks for reviewer**
>
> Dear Reviewer,
>
> We noticed that you haven't yet provided a response, and we appreciate your recognition of our work. Currently, our submission is on the borderline for acceptance. Would it be possible for you to consider increasing one more score? This would greatly helpful for the work.
>
> We are confident in the contribution of our research. The field of hand abstraction has seen little progress in the past decade, and our work offers a seamless application to current poker AI systems, potentially improving their performance significantly. Moreover, our research introduces new ideas that could inspire a series of further advancements in hand abstraction.
>
> Thank you for your consideration.
>
> Best regards,

---

### Official Review · Reviewer_LQq9 · 2024-07-15

**Soundness:** 3
**Presentation:** 1
**Contribution:** 2
**Rating:** 5
**Confidence:** 1

**Summary:**

This paper focuses on the problem of hand abstraction for Texas Hold-Em style poker games. Hand abstraction is the process of partitioning game histories into infosets which still contain enough information to make strategically advantageous decisions. Previous approaches have focused on abstractions that primarily focus on the future outcomes from each hand, but the authors suggest that it may instead be beneficial to also include past information. They design a hand abstraction algorithm called KwrEmd that outperforms previous work by incorporating historical information.

**Strengths:**

The results seem to show that KrwEmd outperforms other imperfect recall hand abstraction algorithms in terms of exploitability.

**Weaknesses:**

As somebody who is unfamiliar with the subfield of imperfect recall abstraction, I found the paper to be quite confusing throughout. The authors do not often provide intuition or examples for their method, and I found it difficult to tell exactly which contributions were novel compared to Fu et al. While it's reasonable for a paper to use technical language at times, well-written papers are usually understandable by a broader set of readers than just those in the specific subfield.

**Questions:**

No specific questions.

**Limitations:**

Not sure.

---

> ### Author Rebuttal · Authors · 2024-08-05
>
> We apologize for the difficulties you experienced while reading my paper, and we appreciate that you did not give a very negative score, giving me the opportunity to further present my work.
>
> Your suggestions are excellent. In the revised version, I will incorporate more intuitive examples (especially examples from Texas Hold'em) to better illustrate the symbols and definitions.
>
> Our writing may have made it difficult for you to grasp the structure of this work, so I will highlight the key points here. This paper builds on the work of Fu et al., who were the first to discover that the mainstream hand abstraction algorithms (signal abstraction algorithms) such as Ehs and PaEmd misused the term "imperfect-recall". In reality, these algorithms follow a "never-recall", "zero-recall", or "forget-everything" paradigm, meaning they abstract hands based solely on the future states and outcomes of the game. This paradigm results in significant information loss. In light of this, they developed the **common refinement** tool and constructed the **k-recall outcome features** and **k-recall outcome isomorphism (KROI)** to demonstrate that considering historical information can greatly facilitate hand abstraction.
>
> However, the work of Fu et al. also has its limitations. Primarily, k-recall outcome features cannot be further clustered (as k-recall outcome features only represent a probability distribution, making it difficult to define the concept of distance), and k-recall outcome isomorphism is too data-intensive to be directly applied in large-scale games like Texas Hold'em. Therefore, there is a need in the community to develop an abstraction algorithm that considers historical information and can be further simplified through clustering. Moreover, since 2014, there have been no updates to applicable hand abstraction algorithms. This is the motivation behind this paper.
>
> We creatively discovered that simplifying k-recall outcome features to k-recall winrate features only reduces a small amount of information, but it allows us to introduce the concept of distance (i.e., the difference in win rates). Based on this discovery, we developed the **k-recall winrate isomorphism (KRWI)** and the **KrwEmd** algorithm. KrwEmd is the first hand abstraction algorithm that considers historical information and can be used in large-scale games. We validated it in the Numeral211 toy environment, and the results show that KrwEmd can significantly improve the performance of poker AIs that originally used Ehs and PaEmd.
>
> It is also important to note that although we have only validated KrwEmd in the Numeral211 environment, this is because the computational cost of evaluating the exploitability metric in larger-scale poker games is quite high. Indeed, KrwEmd can be applied to larger-scale games as well.
>
> Since the poker AIs that caused a sensation between 2016 and 2019 (such as Deepstack, Libratus, and Pluribus) used the PaEmd abstraction algorithm, the PaEmd module can be seamlessly replaced with KrwEmd. This is because hand abstraction is a relatively independent task in the domain of poker AI, with Deepstack, Libratus, and Pluribus primarily focused on strategy training and search. Consequently, the resulting AI could significantly improve performance. This represents a substantial innovation and contribution to the field of poker AI development.

---

> > ### Comment · Reviewer_LQq9 · 2024-08-07
> >
> > Thank you for the clarifications. I think that the paper's presentation could be significantly improved by incorporating some of the explanations given in your rebuttal—this will clarify the significance of the paper to a wider audience.

---

> ### Author Response · Authors · 2024-08-07
> **Thanks for the review**
>
> We are pleased to have had the opportunity to explain our work to you. Thank you for raising your evaluation of our work, which is very important and helpful to us.
>
> May you continue to find success in your work and happiness in your life.
>
> Best regards,
> the authors.

---

### Official Review · Reviewer_2m7b · 2024-07-17

**Soundness:** 2
**Presentation:** 2
**Contribution:** 3
**Rating:** 6
**Confidence:** 1

**Summary:**

This paper proposes KrwEmd, a novel hand abstraction algorithm for imperfect recall settings in Texas Hold'em poker. The algorithm leverages K-recall winrate features, incorporating historical information in addition to future information for constructing hand abstractions. The authors introduce two new isomorphism frameworks: Potential Winrate Isomorphism (PWI) and K-recall Winrate Isomorphism (KRWI). They demonstrate that KRWI outperforms existing methods like POI in identifying distinct infosets. KrwEmd, which combines KRWI with Earth Mover's Distance (EMD) for hand classification, shows superior performance compared to POI, Ehs, and PaEmd in the Numeral211 Hold'em environment.

**Strengths:**

*Originality*: The paper presents a novel combination of K-recall winrate features and EMD for hand abstraction in imperfect recall settings, addressing a critical limitation of current approaches that solely rely on future information.
The introduction of KRWI and PWI provides valuable new tools for understanding and constructing hand abstractions in poker AI.

*Quality*: The experimental results in the Numeral211 environment demonstrate a clear improvement over existing methods, supporting the claims of the paper. The paper includes an appendix with algorithm details and supplementary experimental data.

*Significance*: The proposed KrwEmd algorithm advances the state-of-the-art in hand abstraction for imperfect recall settings, offering a potentially significant improvement for developing stronger poker AI agents.
The incorporation of historical information is a valuable contribution that benefit positively future research in poker and other imperfect information games.

**Weaknesses:**

I found the paper  challenging to understand, though this may be due to my limited background knowledge in poker AI and game theory.  While the authors provide a background section, the density of the technical content and the numerous specialized terms make comprehension difficult.

The description of the accelerated algorithm in Appendix A.3 could be expanded for better understanding. Additionally, a clear discussion of the limitations of the accelerated algorithm would be beneficial.

The paper provides limited information about the proposed algorithms, particularly KrwEmd. While the core concepts are presented, the details regarding implementation and specific design choices are limited. More in-depth explanation and pseudocode would enhance the paper's quality.

**Questions:**

What are the limitations of the accelerated algorithm? Does it introduce any approximations or trade-offs in performance or accuracy?

Have you explored the application of KrwEmd to other poker variants beyond Numeral211? Are there any specific challenges or adaptations needed for different game settings?

How sensitive is KrwEmd to the choice of hyperparameters (w0, w1, w2)? Is the algorithm robust to different hyperparameter configurations?

**Limitations:**

The authors acknowledge the computational complexity of clustering with EMD and introduce an accelerated algorithm. However, the paper lacks a dedicated section addressing the limitations of the proposed methods and the accelerated algorithm.

---

> ### Author Rebuttal · Authors · 2024-08-04
>
> Thank you for your positive evaluation of my work. Your feedback is very constructive and helpful.
>
> To address the weaknesses you pointed out, I will make the following improvements:
>
> You and other reviewers pointed out that my paper is not very easy to read. I acknowledge this issue and appreciate the feedback. Specifically, you mentioned that the high density of background information and the overly abstract and symbolic descriptions contribute to the difficulty. To address this, I will include some examples in the appendix to clarify which components of the Texas Hold'em game each symbol corresponds to.
>
> I will add an intuitive description of the acceleration algorithm (mainly because the distance calculations for k-POI classes given a centroid with all points are repeatedly applied, with the acceleration algorithm avoiding redundant EMD distance calculations). Additionally, I will include a case study demonstrating how the distance calculation for a centroid to all inputs is reduced in the acceleration algorithm, and provide a more accurate complexity comparison. Regarding the limitations of the acceleration algorithm, there are some considerations I need to mention. We noted that this acceleration algorithm is designed to run in a distributed environment (although it can also run in a non-distributed environment, it allows for distributed splitting in centroids). In a distributed environment, the algorithm cannot globally select the most suitable top k initial centroids during the initial centroids selection process. You mentioned a similar issue in question section, and I will discuss this point further later. However, apart from the initial centroids selection process, the results of the clustering process of the acceleration algorithm are equivalent to those of the non-accelerated algorithm, whether in a distributed or non-distributed environment. In a non-distributed environment, the acceleration algorithm is completely equivalent to the non-accelerated algorithm.
>
> Your suggestions on the algorithm description are very important, and we will take them seriously. In the revised version, I will provide a more readable algorithm description. Additionally, if you have any specific suggestions for improvements, I would greatly appreciate them.
>
> Answer to the questions:
>
> - What are the limitations of the accelerated algorithm? Does it introduce any approximations or trade-offs in performance or accuracy?
>
>   This acceleration algorithm does not introduce approximations; its behavior in distance calculations and clustering is identical to the original Kmeans++ algorithm. The acceleration algorithm makes a compromise in the selection of initial centroids compared to the Kmeans++ algorithm (however, this issue exists in almost all distributed Kmeans++ algorithms, which is another problem). Our approach to this scenario is for each distributed node (a total of m nodes) to provide (k/m) initial centroids. When confirming a new local centroid, after updating the minimum distances from the points in each local dataset to the selected local centroids, we use these distances with the softmax operator to probabilistically select the next centroid, instead of simply choosing the point with the largest minimum distance as the new centroid. The benefits of this approach are evident.
>
> - Have you explored the application of KrwEmd to other poker variants beyond Numeral211? Are there any specific challenges or adaptations needed for different game settings?
>
>   The choice of this environment was based on the following considerations: Previously, the toy environment was Flop Hold'em (which limits two-player limit Texas Hold'em to 2 phases). The issue with this environment is that with only two rounds, it cannot reflect the deficiencies of POI abstraction (the number of information sets has a spindle-shaped rather than triangular distribution). Therefore, we needed an environment with at least 3 phases.
>
>   The next problem is that in environments with at least 3 phases, the computational cost of calculating the best response is very high. To create a graph showing exploitability changes with training epochs, we needed a simpler environment. Initially, we chose a game with a deal sequence of [2,1,1] using a standard deck, but the best response calculation was still slow in this environment. One variant of poker is the Royal Deck (which only includes [T, J, Q, K, A] × [♠, ♣, ♥, ♦], a total of twenty cards). However, with this setup and a deal sequence of [2,1,1], the number of recognizable LI is very small, making k-means clustering less convincing. Therefore, we designed a custom environment with 40 cards. This environment is ideal in terms of best response calculation time and the number of recognizable LIs.
>
> - How sensitive is KrwEmd to the choice of hyperparameters (w0, w1, w2)? Is the algorithm robust to different hyperparameter configurations?
>
>   We have not explored this hyperparameter extensively, primarily because each experiment takes about a week more, limiting our exploration. The current conclusion is that setting (w0 > w1 > w2) yields better results, indicating that information from later phases is more important than information from earlier phases. However, further research is needed to determine the optimal hyperparameter settings.
>
> Regarding the limitation section:
>
> The main limitation of our algorithm actually comes from the scale of Texas Hold'em (the data volume is extremely large). However, in a distributed environment, we can utilize several weeks to obtain an abstraction (our distributed environment consists of a cluster of 10 servers, each with 96 logical cores). Although this computational cost is indeed very high, it is also acceptable because, in the process of constructing a high-performance Texas Hold'em AI, calculating the abstraction data is a one-time data preprocessing task. Spending over a month to obtain high-quality abstraction data is worthwhile.

---

### Official Review · Reviewer_uY4u · 2024-07-23

**Soundness:** 2
**Presentation:** 2
**Contribution:** 2
**Rating:** 5
**Confidence:** 2

**Summary:**

The paper develops new hand abstraction techniques for Texas Hold'em-style games (in general: games with ordered signals), which fare better than previous methods in both the number of hands identified, and performance (exploitability) in a simplified version of the game.

Hand abstraction is a technique aiding the strategy construction in a Texas Hold'em, where concrete hands, or rather concrete signal infosets (i.e. "possible words" according to the information revealed so far), are replaced by abstract infosets, represented in an abstract feature space (here $\mathbb{R}^n$).

The core idea of the paper is to use the features of hands from previous rounds in the construction of the current round feature. More precisely, the paper investigates a simple method (KRWI = *k-recall winrate isomorphism*) of maintaining, at a given round, the collection of all potential-winrate isomorphims (PWI) features from previous $k$ rounds by concatenating them all together. PWI for an $n$-player game is a categorical probability distribution over $n+1$ events of a form *"this player outperformed exactly $l-1$ other players while losing to none"* for $l = 1, 2, \ldots n$ and *"this player lost to at least one player"*. Those distributions can be computed by a dynamic programming method. To reduce the cardinality of the space, the paper later clusters KRWI features with k-means using the Wasserstein distance, naming it the KrwEmd method.

All of the methods are benchmarked against currently used techniques that do not use historical information. Experiments find that KRWI identifies similar proportion of signal infosets as the previously used KROI. Using the metric of exploitability of the equilibrium (how it deviates from Nash equilibrium) of the strategy found by an imperfect-information game solver, authors find that 2-RWI-based approach performs almost the same as 2-ROI, and that KrwEmd outpefrorms previously used Ehs and PaEmd by a relatively large margin.

**Strengths:**

The paper proposes a reasonable extension of the currently used techniques for hand abstraction in Texas Hold'e, and shows that the new idea beats SOTA. It does not shy away from introducing the reader to the relevant background in a rigorous way (which is what made it possible for me to even start reading it). Experiments show a meaningful improvement, and provide additional insights (such as the decreasing worth of historical information).

**Weaknesses:**

From the perspective of someone not at all acquainted with the field of imperfect information games/games with ordered signals, the paper was quite hard to read and understand - even though (assuming that the authors agree with my summary), the contribution is a relatively straightforward idea.

The introduction was uninformative and confusing (I would recommend rewriting the whole second paragraph); the preliminaries, although presented in-depth and trying to be formal, also posed quite a few questions; sections 4 and 5 describing the main contribution lacked detail and justification (i.e. ideally I would like to see definition/theorem/proof style - otherwise the text is impossible to read for someone unfamiliar with the field), and the experimental setup is assuming a lot of background knowledge that was not explained neither in the main paper nor in the appendix (it was also difficult to gauge if the comparison between SOTA and the new approach was fair from the resources pov - the paper reports some numbers, but never an aggregated "memory/time used" for all methods).

Please see Questions below for a detailed explanation of what I found lacking or hard to understand.

**Questions:**

Intro:
- What are the "clustering settings" referred to in line 66? (It's difficult to understand how important it is for the claim 'outperforms the SOTA')

Prelims:
- I think the definition of the game $\Gamma$ should include the initial distribution over signals, to start iterating the map $\varsigma
: \Theta \to \Delta(\Theta)$
- I did not understand how exactly the map $\varsigma$ interacts with the shape of the game tree given by $X, \tau$ and the order $\sqsubseteq$ on $\Theta$ - are we assuming that $\theta \sqsubseteq \theta'$ for all $\theta'$ in the codomain of $\varsigma(\theta)$ (dividing the signal space appropriately)? Are the terminal signals $\tilde{\Theta}$ take wrt to the order $\sqsubseteq$, or wrt to the "realisable" order of '$\theta$ is final if it can only ever appear as the final signal of any game trajectory'? Is the order $\sqsubseteq$ assumed to be tree-shaped? (line 12 of algorithm A2 requires a unique predecessor of $\vartheta$)
- Why is the order $\preccurlyeq$ assumed to be partial? Shouldn't it be total on $\tilde{\Theta}$?
- Are the survival status and the $\preccurlyeq$ relevant to the paper? Is the signal abstraction refinement relevant to the paper?
- Why is $\gamma$ provided in the definition of the game and not just derived from other structures?
- What happens to observations $O$ and the map $\varsigma$ in the signal abstracted game $\Gamma^{\alpha)?
- Line 121 - "such that the union of those" should be just "such that they"
- Is the criterion of the signal perfect recall for a game $\Gamma$ equivalent to saying that "players can be Markov"? I.e. that for any non-Markov (wrt to signals) strategy profile $\pi = (\pi_i, \pi_{-i}$) there exists a Markov (wrt to signals) strategy $\pi'_i$ such that $(\pi'_i, \pi_{-i})$ is equivalent to $\pi$?
- I think the signal space $\Theta$ has to be assume to be finite - otherwise, the definition of perfect recall breaks.

Winrate Isomorphism:
- What are the outcome-based features? (never explained, but used throughout this and next sections). What is POI?
- Line 179 - "an identical Winrate-based feature uniquely determines an abstracted signal infoset" - is that a definition of the abstracted signal infosets for winrate-based features? If so, this requires proof or justification (that they partition nicely, satisfy order conditions, interact with  $\varsigma$ in the right way etc).
- The above question is even more relevant for KrwEmb, which uses a complicated clustering mechanism inside.
- Why is $\mathcal{D}$ an isomorphism?
- Line 196 - typo in "classify"
- What is the intuition behind constructing PWI in this particular way (i.e. with "lose at least to one other player"/"win with with $l-1$ players and tie with the rest")? Is this a natural choice? Why not "lose to $l$ players$" or some other metric?
- What is the River phase? What is HUNL&HULHE?
- It would be good to include at least the definition of the Wasserstein distance, instead of linking to Wikipedia

Experiments:
- What is "mb/g" in Figures 4 and 5?
- The plots do not show LI (potentially overshadowed?) and overshadow 2-RWI. I would recommend changing the style, since this is one of the few most important results in this section.
- Line 299 - "We gauge the performance over exploitability" - what does that mean?
- Line 331 - typo in "infosets"
- Line 336 - "the final number of abstracted infosets is set to" - are these numbers for Ehs, PaEmd and KwrEmd respectively? In what order?
- Line 338 - why does KrwEmd use so many parameters here, instead of just 3?
- If the costs of 1000x1427.7s, 12x11.2s and 96.7x341.4s are total costs, wouldn't it be more fair to set the hyper-parameters of the algorithms such that their total computational budget is approximately equal? Otherwise, it's difficult to judge the performance improvement.

**Limitations:**

Yes.

---

> ### Author Rebuttal · Authors · 2024-08-03
>
> Thank you for providing such a thorough review. Your diligence and responsibility are truly impressive, and I am very grateful for your positive evaluation of my paper and the highly constructive feedback. This is extremely helpful for me to improve my work, regardless of whether my paper is accepted for the conference. I would be honored to further introduce my work to you, answer any questions you may have, and correct the mistakes and inaccuracies you pointed out.
>
> I will first address the issues you pointed out in the **Weaknesses** section.
>
> You and the other reviewers mentioned that my writing tends to be somewhat opaque.  I acknowledge and appreciate your critique. I attempted to use symbolic language to convey the background and methods. The advantage of this approach is that it allows for a relatively rigorous description of algorithms. For example, in my paper, I refer to the "k-recall winrate feature." If described in prose, this might look like, "We characterize the signal based on the results of the player's order with others derived from all terminal signals that can be exported. This feature has n+1 dimensions, representing the probability of n+1 different events, which are categorized as...". While this might generally convey the motivation of the work, it often ends up being ambiguous and confusing in detail. On the other hand, using symbolic language, as long as the reader is willing to spend time studying it, there is a higher likelihood of understanding the detailed algorithm I describe. However, I may have overly relied on this symbolic descriptive style and neglected the readability of the paper. Going forward, I will include more examples and diagrams to assist in the explanation, making the paper easier to read and less likely to deter readers.
>
> In the **Weaknesses** section, you mentioned that the ideas presented in the paper are straightforward, which is correct. I am not sure if you meant that the idea seems somewhat trivial, potentially leading to a perception of lower innovation and contribution. Just to clarify, I would like to address this aspect briefly.
>
> The main issue addressed in this paper is the significant information loss associated with methods like Ehs and PaEmd, as highlighted by Fu et al., due to their failure to utilize historical information[^1]. The community has struggled to provide abstraction methods that incorporate historical information (Fu et al.'s work cannot flexibly adjust the scale of abstraction, making it impractical for real-world applications). While the motivation behind the abstraction algorithm is relatively straightforward, our novel discovery is that winrate-based isomorphism results in less information loss compared to outcome-based isomorphism (we believe this finding is significant, akin to lifting a veil that has obscured the truth). Based on this phenomenon, we developed KrwEmd, the first practical abstraction algorithm that incorporates historical information. Since hand abstraction in the Poker AI community is relatively independent and previous work failed to take the advantage of historical information, our work can systematically improve the performance of Poker AI by replacing these modules, thereby making a comprehensive contribution to the entire field.
>
> You provided many writing suggestions in the **Weaknesses** section, and we will take them seriously and make necessary revisions, including but not limited to: restructuring the entire second paragraph, carefully considering the content of the background knowledge section, and removing or simplifying parts that are less relevant to this paper while enriching parts that are highly relevant. We will add more details in Sections 4 and 5 (or supplement the relevant content in the appendix) and enhance the description of the experimental setup.
>
> Regarding the fairness of computational resources in experimental comparisons, we can confirm that while KrwEmd consumes more computational resources during the preprocessing stage (constructing the KrwEmd abstraction compared to Ehs and PaEmd abstractions), this overhead is a one-time cost and will not recur during actual application. In the evaluation phase (i.e., during the actual AI competition, which is repeated in applications), KrwEmd will not introduce additional overhead. Using KrwEmd will systematically improve the AI's performance, and we will emphasize this point in the revised paper.
>
> Due to the large number of questions you have raised, I am unable to address all of them in this thread. I will answer these questions in the comments, so please check them there.
>
> # References
>
> [^1]: Yanchang Fu, Junge Zhang, Dongdong Bai, Lingyun Zhao, Jialu Song, and Kaiqi Huang. Expanding the resolution boundary of outcome-based imperfect-recall abstraction in games with ordered signals. arXiv preprint arXiv:2403.11486, 2024.
>
> [^2]: Finnegan Southey, Michael Bowling, Bryce Larson, Carmelo Piccione, Neil Burch, Darse Billings, and Chris Rayner. Bayes’ bluff: opponent modelling in poker. In Proceedings of the Twenty-First Conference on Uncertainty in Artificial Intelligence, pages 550–558, 2005.
>
> [^3]: Andrew Gilpin and Tuomas Sandholm. Lossless abstraction of imperfect information games. Journal of the ACM (JACM), 54(5):25–es, 2007.
>
> [^4]: Kevin Waugh. A fast and optimal hand isomorphism algorithm. In AAAI Workshop on Computer Poker and Incomplete Information, 2013.
>
> [^5]: Jiefu Shi and Michael L Littman. Abstraction methods for game theoretic poker. In Computers and Games: Second International Conference, CG 2000 Hamamatsu, Japan, October 26–28 2000 Revised Papers 2, pages 333–345. Springer, 2001.

---

> ### Author Response · Authors · 2024-08-05
> **The explanations of some confusing terms**
>
> Thank you for pointing out the typos, which will be corrected in the revised version. It seems that many of my terms and abbreviations were not clearly explained. In the revised version, I will include a glossary in the appendix. For now, I will provide preliminary explanations for the terms you found confusing.
>
> - What are the outcome-based features? (never explained, but used throughout this and next sections). What is POI?
>
>   In line 164, we expand the abbreviation POI to "potential outcome isomorphism," which originates from the paper [^1]. Outcome-based features is a general term for k-recall outcome features and potential outcome features, introduced in the same paper.
>
>   To explain these concepts, we take the Leduc Hold'em environment[^2] as an example, focusing on potential outcome features and potential outcome isomorphism. Due to space constraints, we will not discuss k-recall outcome features. We denote a player's hand in the first phase as [x] and in the second phase as [x;y], where x is the hole and y is the board. Potential outcome features are equivalent to potential winrate features in the terminal phase. Before the final phase, a potential outcome feature describe a distribution of potential outcome features for the next phase (or, equivalently, a distribution of potential outcome isomorphisms). Potential outcome isomorphism refers to classifying hand categories using potential outcome features.
>
>   For example, in Leduc Hold'em, there are nine possible hand combinations in the final phase (phase 2): [J;J], [J;Q], [J;K], [Q;J], [Q;Q], [Q;K], [K;J], [K;Q], and [K;K]. However, their potential winrate (outcome) features fall into only three categories. Therefore, the potential outcome isomorphism in this phase consists of three classes:
>
>   Table.1
>
>   | POI | hands               | (loss, draw, win) |
>   |:---:|:-------------------:|:-----------------:|
>   | 2-1 | [J;J], [Q;Q], [K,K] | (0, 0, 1)         |
>   | 2-2 | [Q;K], [K;J], [K,Q] | (0.25, 0.25, 0.5) |
>   | 2-3 | [J;Q],[J;K],[Q;J]   | (0.75, 0.25, 0)   |
>
>   In the first phase, there are three possible hand combinations: [J], [Q], and [K]. Their potential outcome features also fall into three distinct categories. Therefore, the potential outcome isomorphism in this phase consists of three classes:
>
>   Table.2
>
>   | POI | hands | (2-1, 2-2, 2-3)                            |
>   |:---:|:-----:|:------------------------------------------:|
>   | 1-1 | [J]   | (0.2 {1[J;J]}, 0, 0.8 {2[J;Q]+2[J;K]})     |
>   | 1-2 | [Q]   | (0.2 {1[Q;Q]}, 0.4 {2[Q;K]}, 0.4 {2[Q;J]}) |
>   | 1-3 | [K]   | (0.2,0.8 {2[K;J]+2[K;Q]},0)                |
>
> - Why is an isomorphism?
>
>   Using the term isomorphism, we aim to express a classification equivalence. For example, in Table.1, the hands [J;J] and [Q;Q] have the same potential outcome (winrate) features, and therefore, they are isomorphic in terms of potential outcome (winrate) features.
>
> - What is the River phase? What is HUNL&HULHE?
>
>   HUNL stands for Heads-Up No-Limit Texas Hold'em, and HULHE stands for Heads-Up Limit Hold'em (see line 225). The rules regarding hands are the same for both HUNL and HULHE, which is why I group them together. In both HUNL and HULHE, the game is divided into four phases: Preflop (phase 1), Flop (phase 2), Turn (phase 3), and River (phase 4).
>
> - What is "mb/g" in Figures 4 and 5?
>
>   We indeed overlooked the explanation of this unit. mb/g means milli-blind per game, which can be understood as the number of blinds won or lost per thousand games. In HUNL and HULHE, a similar unit is mbb/g (milli-big-blind per game). However, in Numeral211 Hold'em, we do not distinguish between small and big blinds.

---

> ### Author Response · Authors · 2024-08-05
> **Answers for the questions**
>
> ## Intro
>
> - What are the "clustering settings" referred to in line 66? (It's difficult to understand how important it is for the claim 'outperforms the SOTA')
>
>   By "clustering settings," we refer to the approach of using clustering techniques for hand abstraction, with representative techniques being Ehs and PaEmd. Although clustering-based abstraction has become mainstream, there are still other approaches, such as the Lossless Isomorphism methods developed by Gilpin and Sandholm and Waugh[^3][^4], as well as rule-based abstraction methods based on human expertise[^5]. This confusion arises from my choice of words; a more accurate expression would be "In the context of clustering-based abstraction algorithms,...".
>
> ## Prelims
>
> - I think the definition of the game  should include the initial distribution over signals, to start iterating the map.
>
>   The distribution of signals is fixed and known to everyone, but you cannot have the opponent's observation of the given signal (opponent's hole), and therefore cannot determine the distribution of subsequent signals. For example, in Leduc Hold'em, if a player receives the hole [J] and knows that the opponent's hole is [Q], they can accurately determine that the probabilities of the board being [-; J], [-; Q], and [-; K] are 0.25, 0.25, and 0.5, respectively. However, since the player cannot know the opponent's hole, they can only estimate a prior probability: the board has a 0.2 probability of being [-; J], a 0.4 probability of being [-; Q], and a 0.4 probability of being [-; K]. This distribution is typically known in poker games, similar to looking up a table, so there is no need to discuss the initial distribution separately.
>
> - I did not understand how exactly the map $\\varsigma$ interacts with the shape of the game tree given by $X, \\tau$ and the order $\\sqsubseteq$ on $\\Theta$ - are we assuming that $\\theta \\sqsubseteq \\theta^{\\prime}$ for all $\\theta^{\\prime}$ in the codomain of $\\varsigma(\\theta)$ (dividing the signal space appropriately)? Are the terminal signals $\\tilde{\\Theta}$ take wrt to the order $\\sqsubseteq$, or wrt to the "realisable" order of ' $\\theta$ is final if it can only ever appear as the final signal of any game trajectory'? Is the order $\\sqsubseteq$ assumed to be tree-shaped? (line 12 of algorithm A2 requires a unique predecessor of $\\vartheta$ )
>
>   Signals can indeed be understood as a tree structure because we require each signal to have a unique predecessor. In Leduc Hold'em, a possible signal is [J, Q; K], which is a second-phase signal indicating that in the first phase, player 1 received the hole J and player 2 received the hole Q, and in the second phase, the board K was dealt. The unique predecessor of this signal is [J, Q], a first-phase signal indicating that player 1 received the hole J and player 2 received the hole Q. Symbolically, we have [J, Q] $\\sqsubseteq$ [J, Q; K].
>
>   $\varsigma$ is only related to the current signal and is independent of $X$ and $\\tau$. As illustrated in the above example, $\\varsigma([J, Q]) = \\{ 0.25[J, Q; J], 0.25[J, Q; Q], 0.5[J, Q; K] \\} $.
>
>   The description of terminal signals is indeed not very detailed, which might confuse readers unfamiliar with poker game rules. We can illustrate this with an example: in Leduc Hold'em, terminal signals are all the second-phase signals, and an order can be defined on these signals. For instance, we have $\\succcurlyeq ([J, Q; K], 1, 2) = false$ and $\\succcurlyeq ([J, Q; K], 2, 1) = true$, meaning that for the terminal signal [J, Q; K], the order of player 1 is lower than that of player 2.
>
>   Are all signals at terminal nodes terminal signals? No. For example, if a player folds in the first phase, it will also create a terminal node, but there is no need to compare orders at this terminal node. Are all terminal signals at terminal nodes? Not necessarily; the terminal signal [J, Q; K] can also appear at internal nodes in the second phase (i.e., when player 1 and player 2 are still making decisions). If we consider signals as a tree structure, we can assert that signals without successors are terminal signals, while those with successors are not. Terminal signals are the leaf nodes of the signal tree.
>
> - Why is the order $\\preccurlyeq$ assumed to be partial? Shouldn't it be total on $\\tilde{\\Theta}$ ?
>
>   You are right, that was a typo. $\\succcurlyeq$ is a total order on terminal signals (i.e. $\\tilde{\\Theta}$), but what I meant is that it is a partial order on $\\Theta$. The domain of $\\succcurlyeq$ projected onto $\\Theta$ is $\\tilde{\\Theta}$ exactly.

---

> ### Author Response · Authors · 2024-08-05
> **Answers for the questions**
>
> - Are the survival status and the  relevant to the paper? Is the signal abstraction refinement relevant to the paper?
>
>   Survival status is part of the definition of ordered signal games. It determines at which nodes the order of signals will affect the player's payoff, thus it is relevant to our discussion. Although it might seem less important when studying signal abstraction alone, omitting this concept might lead to misunderstandings regarding the orthogonal relationship between the signals and public tree.
>
>   Signal abstraction refinement is a relationship between signal abstractions that describes the inclusion relationship in terms of the number of infosets (information sets) identified. In our paper, we use the conclusion that POI is a signal abstraction refinement of the mainstream abstraction algorithms Ehs and PaEmd (see line 311). Applying this conclusion, we only need to demonstrate that if KrwEmd performs better than POI while identifying the same number of information sets as POI, then KrwEmd is a stronger abstraction algorithm than Ehs and PaEmd.
>
> - Why is $\gamma$ provided in the definition of the game and not just derived from other structures?
>
>   In imperfect information games, there is no concept of phases. However, in games with ordered signals, we define phases as the number of times the chance player (nature) takes actions during the game. The definition of phases was first formally described by [^3], and a more detailed definition was provided by [^1]. With the definition of phases, the problem of signal abstraction gains a more mathematical description.
>
> - What happens to observations $O$ and the map $\varsigma$ in the signal abstracted game $\Gamma^{\alpha}$?
>
>   We can use the Table.1 to answer this question. Let's assume $\alpha = (POI, \varTheta_2)$ (i.e., player 1 uses POI abstraction while player 2 does not use any abstraction). $O_2([J,Q;K]) = \\{[J,Q;K], [Q,Q;K], [K,Q;K]\\}$ and $O_1([J,Q;K]) = \\{[J,Q;K], [J,J;K], [J,K,K]\\}\cup \\{[J,J;Q], [J,Q;Q], [J,K;Q]\\} \cup \\{[Q,J;J],[Q,Q;J], [Q,K;J]\\}$。In another representation, we can write $O_2([J,Q;K]) = \\{[-,Q;K]\\}$ and $O_1([J,Q;K]) = \\{[J,-;K]\\}\cup \\{[J,-;Q]\\} \cup \\{[Q,-;J]\\} = POI_{2-3}$.
>
>   The $\varsigma$ in $\tilde{\Gamma}$ and $\tilde{\Gamma}^{\alpha}$ is the same; signal abstraction only affects observation.
>
> - Is the criterion of the signal perfect recall for a game $\Gamma$ equivalent to saying that "players can be Markov"? I.e. that for any non-Markov (wrt to signals) strategy profile $\pi=\left(\pi_i, \pi_{-i}\right)$ there exists a Markov (wrt to signals) strategy $\pi_i^{\prime}$ such that $(\pi^\prime_{i}, \pi_{-i}) \equiv \pi$?
>
>   In imperfect information games, the intuitive meaning of perfect recall is remembering all the information observed by the players throughout the game. In contrast, in stochastic games, the Markov property indicates that the future outcome of the game depends only on the current state (i.e., the current state includes all necessary information). The two concepts can indeed be compared. Regarding the strategy comparison you mentioned, I don't quite understand its specific meaning.
>
> - I think the signal space $\Theta$ has to be assume to be finite - otherwise, the definition of perfect recall breaks.
>
>   You are right. Although we aim to discuss the signal abstraction problem more generally, for now, $\Theta$ should be restricted to a finite set.

---

> ### Author Response · Authors · 2024-08-05
> **Answers for the questions**
>
> ## Winrate Isomorphism
>
> - Line 179 - "an identical Winrate-based feature uniquely determines an abstracted signal infoset" - is that a definition of the abstracted signal infosets for winrate-based features? If so, this requires proof or justification (that they partition nicely, satisfy order conditions, interact with $\varsigma$ in the right way etc).
>
>   We can argue this intuitively, albeit not rigorously. Each signal infoset has a unique potential winrate feature or k-recall winrate feature (obtained by enumerating all opponent holes and then calculating the statistics, a deterministic and non-random process; note that these features are defined on signal infosets, not on individual signals). We then arbitrarily define signal infosets with the same potential winrate feature or k-recall winrate feature as equivalence classes, i.e., an abstracted signal infoset. Therefore, each signal infoset will be in one and only one abstracted signal infoset.
>
> - The above question is even more relevant for KrwEmb, which uses a complicated clustering mechanism inside.
>
>   You can think of it this way: KROI is an input to a K-Means problems, and the K-Means algorithm will assign each input data point to a unique class. In the previous question, we also argued that each signal infoset will definitely be assigned to one of KROI's class. Therefore, KrwEmd will also assign each signal infoset to a unique abstracted signal infoset.
>
> - why is $\mathcal{D}$ an isomorphism
>
>   As mentioned above, we use isomorphism to represent the equivalence with respect to a certain feature. Signal infosets determined to be isomorphic by $\mathcal{D}$ have the same certain feature, such as potential winrate/outcome feature or k-recall winrate/outcome feature.
>
> - What is the intuition behind constructing PWI in this particular way (i.e. with "lose at least to one other player"/"win with with $l-1$ players and tie with the rest")? Is this a natural choice? Why not "lose to $l$ players" or some other metric?
>
>   Since Texas Hold'em is a winner-takes-all game, during the showdown, if you lose to any other player who reaches the showdown, you will lose your chips (though it may not be all your chips due to the all-in and side pot rules, but we are not discussing such complexities here. In academic discussions, the simplified problem typically assumes that all players bring the same amount of usable chips to the game. In this case, if your hand is lower than that of any other player at the showdown, you will lose all your chips). Therefore, as long as your hand is not the highest, it means you lose. A tie occurs only when a player's hand is the highest but not the sole highest hand.
>
> ## Experiments
>
> - Line 299 - "We gauge the performance over exploitability" - what does that mean?
>
>   We use the metric of exploitability to evaluate the performance of each signal abstraction algorithm.
>
> - Line 336 - "the final number of abstracted infosets is set to" - are these numbers for Ehs, PaEmd and KwrEmd respectively? In what order?
>
>   The word "final" is a typo that was mistakenly added and it introduced ambiguity. (100, 225, 396) As described in line 320, phase 1 sets 100 abstracted signal information sets, phase 2 sets 225 abstracted signal information sets, and phase 3 sets 396 abstracted signal infosets.
>
> - Line 338 - why does KrwEmd use so many parameters here, instead of just 3?
>
>   We apply KrwEmd in the Numeral211 Hold'em environment. Clustering is not needed in the first phase due to the small number of singal infosets, so we directly use LI. We cluster the signal infosets for the second and third phases separately. Clustering in the second phase needs to consider two phases, phase 1 and phase 2, requiring two parameters $(w_{2,0}, w_{2,1})$. Clustering in the third phase needs to consider three phases, phase 1, phase 2, and phase 3, requiring three parameters $(w_{3,0}, w_{3,1}, w_{3,2})$. A total of five parameters are needed.
>
> - If the costs of 1000x1427.7s, 12x11.2s and 96.7x341.4s are total costs, wouldn't it be more fair to set the hyper-parameters of the algorithms such that their total computational budget is approximately equal? Otherwise, it's difficult to judge the performance improvement.
>
>   As we mentioned in the **Weakness** section, signal abstraction is a data preprocessing process that is computed only once in the entire Poker AI construction. As long as it is computable, even if it takes months to compute high-quality hand abstraction data, it is worth the effort. These hand abstraction data can then be used for AI iterations with algorithms such as CFR, as well as for real-time strategy solving. We acknowledge that our algorithm has a very high computational cost, mainly because the input data considering historical information is several orders of magnitude larger than the input data that ignores historical information. However, this results in much higher quality abstraction data compared to previous algorithms.

---

> > ### Comment · Reviewer_uY4u · 2024-08-12
> > **Response**
> >
> > I thank the authors for their thorough response and addressing all of the points raised in my review. I think the paper can be significantly enhanced by incorporating those explanations into the text, as promised by the authors.
> >
> > Two minor points regarding the above:
> >  - In the question about $\gamma$, I meant that, as far as I understand, it is possible to deduce it from $\rho$ and $\tau$ - so I am puzzled as to why it is (seemingly unnecessarily) included in the game definition.
> >  - Even though for poker specifically, the initial distribution is known, an abstract definition of a game with imperfect information still requires it to be specified.
> >
> > I decided to keep my original rating.

---

> > > ### Author Response · Authors · 2024-08-12
> > > **Thanks for review**
> > >
> > > During our discussion, we identified another point that could enhance the rigor of the paper, specifically related to your question about $\gamma$.
> > >
> > > $\gamma$ is indispensable in the definition, as it is one of the key concepts that distinguishes imperfect information games from sequential signaling games. Our work on hand abstraction is also based on $\gamma$ in games with ordered signals. However, your point is also valid—$\gamma$ can indeed be deduced from $\tilde{\rho}$ and $\tilde{\tau}$ (in fact, its definition originates from this). The issue lies in whether $\gamma$ appears in other components of the definition of $\tilde{\Gamma}$. If it does not, then it could be excluded from the definition and treated as an extended concept. However, if $\gamma$ does appear in other components of $\tilde{\Gamma}$, then it should be included in $\tilde{\Gamma}$. Our original intention was the latter, but this may not have been clearly reflected in the current version. We are considering adjusting the definition of $\Theta$ to $\Theta = (\Theta^{(1)}, \dots, \Theta^{(\max{\mathfrak{r}})})$, so that $\gamma$ would appear in other parts of the definition of $\tilde{\Gamma}$. This adjustment would also allow us to avoid the need for an additional definition of the terminal signal set $\tilde{\Theta}$, as we could directly use $\Theta^{(\max \mathfrak{r})}$ (we have always felt that introducing $\tilde{\Theta}$ in the introduction was somewhat inelegant).
> > >
> > > Regarding your question about the initial state, after careful consideration, we agree that your point is well-founded. Introducing an initial state does enhance the generalizability of the model, allowing it to be applied to different initial states in poker games. For Texas Hold'em, we can define $x^o = (\tilde{x}^o, \phi)$, where $\tilde{x}^o$ is the root public node, and $\phi$ indicates that no cards have been dealt.
> > >
> > > Once again, we sincerely appreciate your thorough and detailed review. Your suggestions have prompted us to think more deeply and improve our paper. Furthermore, our paper is currently in a borderline state with a reasonable chance of acceptance. If you could give us one more score, it would be greatly helpful. We are confident in the contributions of this work, as it could significantly enhance the performance of AI in Texas Hold'em (after all, previous hand abstraction methods have resulted in significant information loss). If this paper could be accepted sooner, it would promptly impact the field (since work in hand abstraction has stagnated for the past decade). We sincerely hope you can lend us your support.
> > >
> > > Best regards,
> > > the authors

---

> > > > ### Comment · Reviewer_uY4u · 2024-08-12
> > > > **Response**
> > > >
> > > > I thank the authors for continued discussion, and I appreciate the willingness to fix the problems. Their answers to my minor points above seem reasonable.
> > > >
> > > > I haven't increased my score mostly because the number of changes (mentioned in response to this review, as well as in other comments) require a substantial re-write of the paper. This seems to be a consensus among the reviewers. It is difficult to judge how will it look like after that, and thus impossible to assign a higher score based on this.

---

> > > > > ### Author Response · Authors · 2024-08-12
> > > > > **Response for the comment**
> > > > >
> > > > > Thanks for your review. It is very helpful. : )

---

### Decision · Program_Chairs · 2024-09-25

**Decision:**

Reject

**Comment:**

This paper proposes a new approach to hand abstraction in Texas Hold'em-style poker games, addressing the limitation of previous methods that disregard historical data. The authors present two novel methods that explore a promising direction to alleviate the existing issue. However, significant improvements are needed to take place before publication. The reviewers' low confidence score and numerous clarification questions on the setting are indication to this. That is, a substantial editing is required to take place before the paper is ready for publication.

I do encourage the authors to use the feedback to improve the work and resubmit it in the future, as it has value for the research community.